# Alpha- and beta-adrenergic octopamine receptors in muscle and heart are required for *Drosophila* exercise adaptations

**Alyson Sujkowski**[1], **Anna Gretzinger**[2], **Nicolette Soave**[1], **Sokol V. Todi**[2], **Robert Wessells**[1]*

**1** Department of Physiology, Wayne State University School of Medicine, Detroit, Michigan, United States of America, **2** Department of Pharmacology, Wayne State University School of Medicine, Detroit, Michigan, United States of America

* rwessell@med.wayne.edu

**Data Availability Statement:** All relevant data are within the manuscript and its supporting information files.

## Abstract

Endurance exercise has broadly protective effects across organisms, increasing metabolic fitness and reducing incidence of several age-related diseases. *Drosophila* has emerged as a useful model for studying changes induced by chronic endurance exercise, as exercising flies experience improvements to various aspects of fitness at the cellular, organ and organismal level. The activity of octopaminergic neurons is sufficient to induce the conserved cellular and physiological changes seen following endurance training. All 4 octopamine receptors are required in at least one target tissue, but only one, *Octβ1R*, is required for all of them. Here, we perform tissue- and adult-specific knockdown of alpha- and beta-adrenergic octopamine receptors in several target tissues. We find that reduced expression of *Octβ1R* in adult muscles abolishes exercise-induced improvements in endurance, climbing speed, flight, cardiac performance and fat-body catabolism in male *Drosophila*. Importantly, *Octβ1R* and OAMB expression in the heart is also required cell-nonautonomously for adaptations in other tissues, such as skeletal muscles in legs and adult fat body. These findings indicate that activation of distinct octopamine receptors in skeletal and cardiac muscle are required for *Drosophila* exercise adaptations, and suggest that cell non-autonomous factors downstream of octopaminergic activation play a key role.

## Author summary

Repeated endurance exercise produces gradual adaptations in animals and humans that improve many aspects of health. One important component of the exercise response is adrenergic signaling. Norepinephrine secretion is known to be upregulated during an exercise bout, and affects multiple organ systems. We previously demonstrated that activation of neurons that secrete the invertebrate version of norepinephrine, octopamine, are necessary for fruit flies to respond to chronic exercise by increasing their endurance, speed and cardiac performance. Here, we use tissue-specific genetics to show which octopamine receptors are required in each tissue for exercise to exert its positive effects on

**Funding:** Funding for this work was provided by R01NS086778 to SVT, NIH 1RO1AG059683 to R. W. and by a Physiology Department Summer Research Fellowship Award to N.S www.med. wayne.edu/physiology/SURF. The funders had no role in study design, data collection and analysis, decision to publish, or preparation of the manuscript.

**Competing interests:** The authors have declared that no competing intersts exist.

long-term health. We find that all the various receptors are required in at least one organ system, and that one receptor, Octβ1R, is required in all of them. We also report that, despite the high similarity between the various receptors, they have distinct responses, and cannot always substitute for one another. We also find that activation of octopamine receptors in skeletal or cardiac muscle have tissue-non-autonomous effects on adaptations in other tissues. These results help to further understand the complex interplay between neuronal signaling and responses in various organs during chronic exercise.

## Introduction

Endurance exercise is a potent, low-cost intervention with broad healthspan-extending effects [1]. Chronic endurance training simultaneously promotes healthy physiology and prevents disease, improving function in heart, skeletal muscle and brain while reducing obesity, heart disease and cognitive decline [2–4]. These benefits are associated with adaptive changes to gene expression and metabolism [2,5–8].

Nonetheless, these benefits are inaccessible to much of the population that are unable to perform an endurance exercise regimen because of injury, illness, advanced age, or lifestyle. Dissecting the mechanisms underlying exercise adaptations in order to identify exercise mimetics remains a prominent research goal. Until recently, studies of life-long exercise effects were limited to rodent models and retroactive comparisons of human cohorts, making controlled, longitudinal analysis and large-scale genetic studies difficult. We and others have developed endurance training programs for *Drosophila*, taking advantage of their innate instinct for negative geotaxis, allowing for controlled training of large, genetically identical cohorts [9,10]. After our 3 week training protocol, male flies increase climbing speed, cardiac stress resistance [11], endurance [12], flight performance [5], and lysosomal activity in their fat body [7]. Trained male flies have increased mitophagy in cardiac and skeletal muscle [13], increased mitochondrial enzyme activity [11,14] and changes in transcript expression similar to those found in long-lived flies [5]. These genetic and physiological adaptations closely resemble benefits seen in both rodent models [15] and humans [1].

We have previously found that increased activity of octopaminergic neurons is both necessary and sufficient for exercise adaptations in *Drosophila*, even in sedentary flies where mobility is restricted by a foam stopper placed low in the vial [8]. Octopamine signals through conserved α and β-adrenergic receptors [16,17], similar to those that bind vertebrate norepinephrine. Both norepinephrine and octopamine have been associated strongly with the fight-or-flight response. In addition to increasing the drive to exercise, octopamine and norepinephrine are known to facilitate transient increases in endurance, lipolysis, and fatty acid metabolism [18–23]. As either daily exercise or daily short-term activation of octopaminergic neurons is sufficient to provide the benefits of endurance exercise in *Drosophila* [8], and chronic exercise stimulates increased norepinephrine secretion in humans [24], these mechanisms appear to be conserved.

Norepinephrine is produced neuronally and also released into circulation from the adrenal gland [25,26]. Its signal can be received by a family of alpha- and beta- adrenergic receptors that are expressed in complementary patterns [27]. The specific requirements for each of these receptors in various tissues in driving the response to chronic exercise is incompletely understood in any organism. While *Drosophila* do not have an adrenal gland, neuronally produced octopamine can signal to neighboring cells or be released into circulation, where it can potentially bind receptors in a variety of tissues [17]. Here, we utilize the *Drosophila* system to

uncover tissue-specific requirements for each receptor in executing the adaptive response to chronic exercise.

*Drosophila* have one α-adrenergic octopamine receptor, OAMB, and 3 dedicated β-adrenergic octopamine receptors, Octβ1R, Octβ2R, and Octβ3R [17]. We have previously established that all 4 are required for at least some aspect of exercise adaptation in *Drosophila*, but only one, Octβ2R, was required for all of the characteristic adaptations when knocked down using global, inducible RNAi [8]. These results are consistent with studies in which β-blockers tend to inhibit exercise performance and adaptation in humans [28,29].

Here, we separately map receptor requirements for exercise adaptations using the UAS--Gal4 system and Gene Switch Gal4 to reduce adrenergic receptor expression in heart, adult fat body and adult skeletal muscle. We find that Octβ2R expression is required in skeletal muscle for improvements to endurance and speed, but also cardiac performance and fat-body autophagy, suggesting tissue non-autonomous effects. OAMB and Octβ3R have tissue-specific effects on exercise adaptations, with OAMB being more important for cardiac improvements and Octβ3R essential for flight. Intriguingly, we also find cell non-autonomous effects resulting from reductions in cardiac-specific OAMB and Octβ1R knockdown, as well as muscle-specific OAMB and Octβ1R knockdown.

## Results

### Octopamine receptor expression is tissue-specific

*OAMB*, *Octβ1R*, *Octβ2R* and *Octβ3R* transcript expression was measured in heart, muscle and fat body using qRT-PCR, in agreement with previously published reports [17] (S1A–S1C Fig). All 4 receptors were detected in adult muscle (S1B Fig), while only *Octβ3R* transcript expression was found in adult fat body (S1C Fig). We also detected for the first time *OAMB* and *Octβ1R* in hearts, supported by RT-PCR and subsequent physiological analyses. *Octβ2R* and *Octβ3R* transcripts were not detected in heart tissue (S1A Fig). Because skeletal muscle, heart and adipose tissue are the primary tissues where we have characterized changes following chronic exercise, and because OA receptors are expressed in each of these tissues, we set out to map which receptors are required in each of these key tissues to execute the effects of chronic exercise training.

We first tested knockdown efficiencies for each RNAi construct used with each driver, separately testing at 72 hours after induction by addition of RU486 (see methods), and again at 25 days after induction (S1A–S1C Fig). Unexercised and exercised cohorts were also separately measured. Knockdowns ranged from 50% to 95% and in most cases were more efficient at 25 days then at 72 hours. There was no consistent effect of exercise on knockdown efficiency.

### OAMB is required in muscle and heart for improved climbing speed, endurance and cardioprotection with exercise

Runspan, a measure of endurance in which *Drosophila* time to fatigue is scored in real time and plotted similarly to a survival curve (see methods), was scored on day 5 post-eclosion, 72 hours after induction of RNAi expression through RU486 feeding. RNAi against *OAMB* (Fig 1) in neither adult muscles (MHC GS RU+, Fig 1A) nor adult hearts (hand GS RU+, Fig 1G) altered baseline endurance. The same cohort of flies was given 3 weeks of exercise training and assessed again for endurance on day 25. Typically, exercise-trained wild-type flies run longer than genetically identical, untrained siblings that have been placed on the machine on each day, but with a foam stopper to prevent climbing as a control for the exercise environment [9]. Genetic background controls were genetically identical but lacked the inducing drug for the

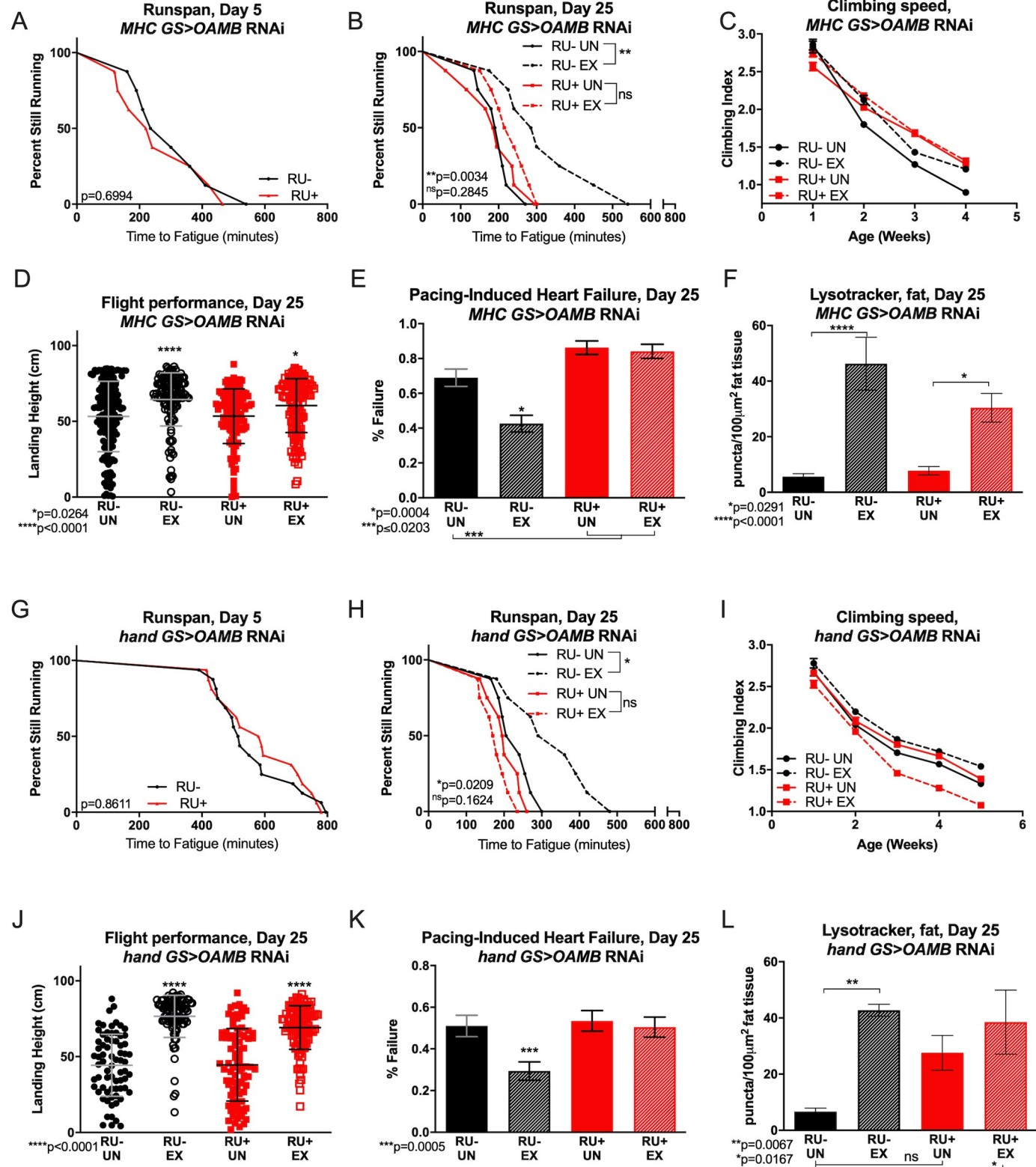

**Fig 1. OAMB is required in muscle and heart for exercise adaptations to climbing speed, endurance and heart performance. (A)** Baseline endurance was not affected by knockdown of *OAMB* in muscle (log-rank, p = 0.6994). **(B)** *MHC GS>OAMB* RNAi RU+ flies do not improve endurance after exercise training (log-rank, p = 0.2845) and have similar endurance to RU- untrained flies (log-rank, p = 0.1508). *MHC GS>OAMB* RNAi RU- exercised have significantly better

endurance than untrained RU- siblings (log-rank, p = 0.0034). **(C)** *MHC GS>OAMB* RNAi RU+ flies do not improve climbing speed with exercise training (2-way ANOVA, exercise effect, p≥0.0620) but have statistically higher climbing index than trained RU- controls at week 3 (2-way ANOVA, genotype effect, p = 0.0002). *MHC GS>OAMB* RNAi RU- flies respond to exercise with increased climbing speed at weeks 2–4 (2-way ANOVA, exercise effect, p≤0.0489). **(D)** Uninduced RU- *MHC GS>OAMB* RNAi control flies respond to exercise with enhanced flight performance (ANOVA with Tukey post-hoc, p<0.0001), as do *MHC GS>OAMB* RNAi RU+ flies (ANOVA with Tukey post-hoc, p = 0.0264). **(E)** *MHC GS>OAMB* RNAi RU+ flies have higher cardiac failure in response to external electrical pacing compared to untrained RU- controls whether exercised or not (Chi-squared, p≤0.0203). RU- background controls improve cardiac performance with exercise with reduced failure rate (Chi-squared, p = 0.0004). **(F)** Both RU+ and RU- *MHC GS>OAMB* RNAi flies increase fat body lysosomal activity after exercise training compared to unexercised cohorts (ANOVA with Tukey post-hoc, p = 0.0291, p<0.0001). **(G)** *hand GS>OAMB* RNAi RU+ and RU- flies have similar endurance at adult day 5 (log-rank, p = 0.8611). **(H)** *hand GS>OAMB* RNAi RU+ flies do not improve endurance after exercise training (log-rank, p = 0.1624) while exercised uninduced RU- controls have better endurance than untrained siblings (log-rank, p = 0.0209). **(I)** Untrained *hand GS>OAMB* RNAi RU+ flies have similar climbing speed to untrained, uninduced RU- controls (2-way ANOVA, genotype effect, p≥ 0.1648) but have reduced climbing speed across ages with exercise (2-way ANOVA, exercise effect, p<0.0001 after week 2). RU- controls improve climbing speed with exercise training (2-way ANOVA, exercise effect, p<0.0001 after week 1). **(J)** Both *hand GS>OAMB* RNAi RU+ flies and RU- controls improve flight performance, measured by landing height, after exercise training (ANOVA, p<0.001). **(K)** *hand GS>OAMB* RNAi RU- control flies respond to endurance exercise with lower cardiac failure in response to external electrical pacing, but RU+ flies do not (Chi-squared test, p = 0.005, p = 0.7169). **(L)** Exercised *hand GS>OAMB* RNAi RU+ flies have increased fat body LysoTracker staining compared to untrained, uninduced RU- controls, although levels are not significantly increased in comparison to untrained RU+ cohorts (ANOVA with Tukey multiple comparisons, p = 0.0167, p = 0.1537, respectively).

RNAi (RU-). Flies with muscle- or heart- specific *OAMB* RNAi failed to increase endurance with exercise, while control flies responded to exercise with increased endurance as normal (Fig 1B and 1H).

Wild-type exercise-trained males retain greater negative geotaxis climbing speed across ages than age-matched control siblings [11]. We assessed negative geotaxis speed 5 times per week prior to the start of daily training, as described [30]. RNAi against *OAMB* in adult muscles prevented adaptation in trained flies, while RU- controls improved normally with exercise (Fig 1C). Heart-specific *OAMB* knockdown more severely impaired climbing speed, with exercised flies actually climbing slower than unexercised siblings. RU- control flies responded to exercise normally (Fig 1I).

Exercise training is cardio-protective in wild-type males, as measured by response to external electrical pacing stress [31]. Both muscle- and heart-specific *OAMB* knockdown prevented cardioprotective benefits of exercise training, while RU- controls responded normally to exercise (Fig 1E and 1K).

Exercise-trained wild-type male flies also have better flight performance as measured by recording landing height after ejection from a platform [8], and wild-type male flies increase autophagy in the fat body [8] during chronic exercise. Knockdown of *OAMB* in muscle or heart did not prevent exercise from increasing flight ability (Fig 1D and 1J) and fat body Lyso-Tracker staining (Fig 1F and 1L and S9A and S9B Fig).

## Octopamine β1 receptor is specifically required in muscle for adaptive response to chronic exercise

Muscle-specific knockdown of *Octβ1R* (MHC GS Octβ1R RU+) reduced baseline endurance in comparison to uninduced controls (Fig 2A). Importantly, *Octβ1R* reduction did not prevent repetitive climbing exceeding 400 minutes, meaning that it did not preclude these flies from performing our exercise regimen. Muscle-specific *Octβ1R* RNAi completely prevented exercise from increasing endurance, and *Octβ1R* RNAi flies ran shorter than untrained RU- controls whether exercised or not (Fig 2B). Octβ1R was also required in muscles for exercise-dependent improvements in climbing speed, flight, and cardiac performance (Fig 2C–2E). Fat body Lyso-Tracker staining was abnormal in muscle-specific *Octβ1R* RNAi flies, with high lysosomal activity whether exercised or not, and unexercised flies actually showing higher activity (Fig 2F, S9C Fig). Heart-specific knockdown of *Octβ1R* (hand GS Octβ1R RU+) had similar detrimental effects on baseline and post-training runspan, climbing speed, and LysoTracker staining, with exercised RU+ flies all performing similar to or worse than untrained, uninduced

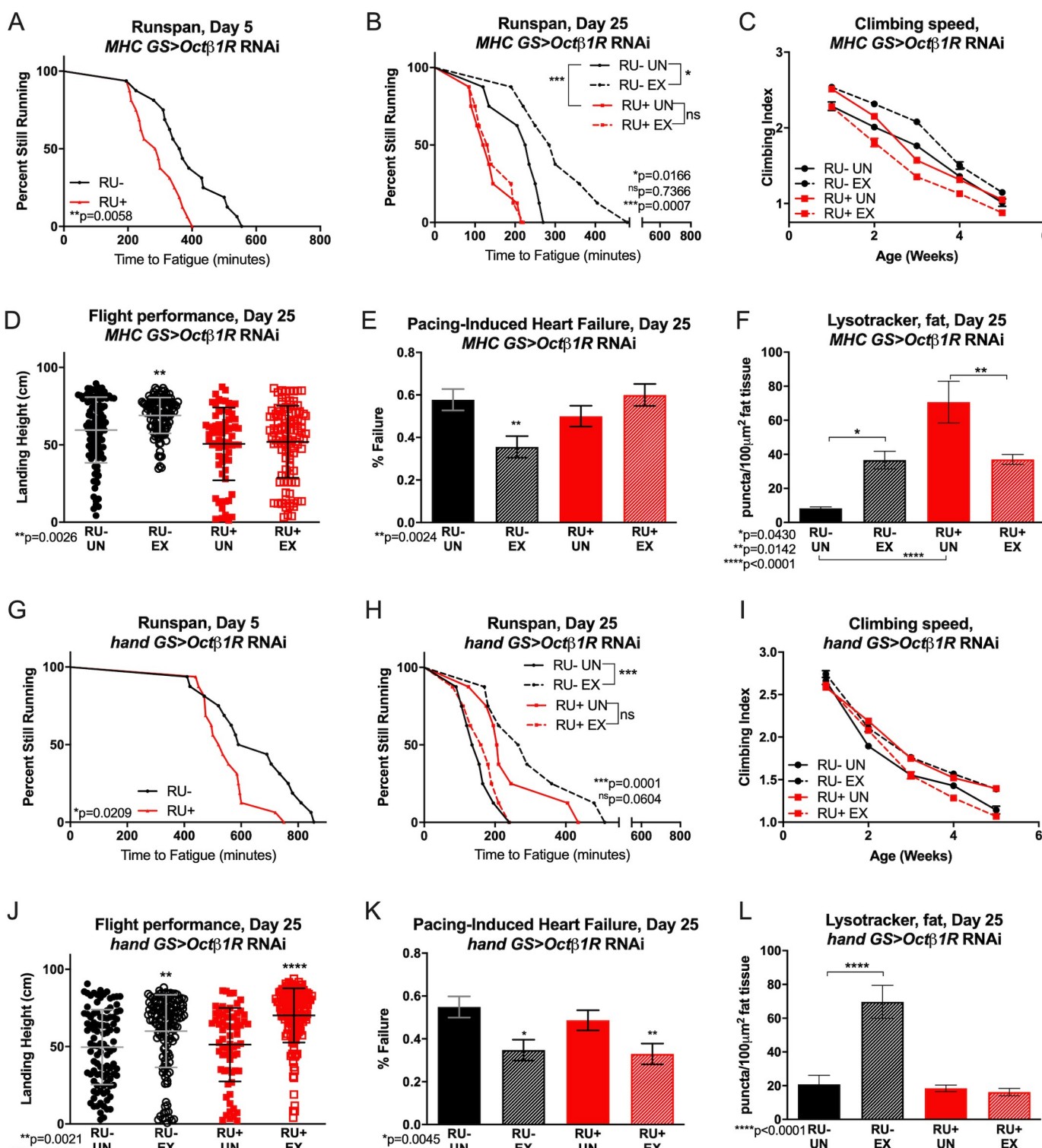

**Fig 2. Exercise adaptations in *Drosophila* require Octopamine *β1* receptors in muscle. (A)** *MHC GS> Octβ1R* RNAi have lower baseline endurance than age-matched RU- control flies (log-rank, p = 0.0058). **(B)** Following exercise, *MHC GS> Octβ1R* RNAi flies have reduced endurance compared to untrained RU-control flies (log-rank, p = 0.0007) and do not improve with training (p = 0.7366). Trained RU- control flies improve endurance compared to untrained siblings (p = 0.0166). **(C)** Exercised *MHC GS> Octβ1R* RNAi RU+ flies climb slower than unexercised RU- flies across ages while uninduced RU- flies have faster climbing speed (2-way ANOVA, exercise effect, p<0.0001). **(D)** In an acute flight performance test, *MHC GS>Octβ1R* RNAi RU+ flies do not improve landing height after exercise. Exercised, uninduced RU- controls have higher landing height (ANOVA, p = 0.0026). **(E)** *MHC GS> Octβ1R* RNAi RU+ flies do not reduce pacing-

induced heart failure after exercise (Chi-squared, p = 0.0752). **(F)** Unexercised *MHC GS> Octβ1R* RNAi RU+ flies have high fat body LysoTracker staining that is reduced following exercise (ANOVA with Tukey multiple comparisons, p = 0.0142). Uninduced, unexercised RU- controls have low fat body lysotracker activity that increases with exercise (ANOVA with Tukey multiple comparisons, p = 0.0430). **(G)** *hand GS>Octβ1R* RNAi RU+ flies have lower day 5 endurance than uninduced RU- controls (log-rank, p = 0.0209), and **(H)** *hand GS> Octβ1R* RNAi RU+ flies do not improve endurance after exercise training (log-rank, p = 0.0604). Trained RU- control flies have better endurance than untrained siblings (log-rank, p = 0.0001). **(I)** Exercise-trained *hand GS>Octβ1R* RNAi RU+ flies have lower climbing speed than untrained siblings after week 3 (2-way ANOVA, exercise effect, p≤0.0011). **(J)** Exercise improves landing height in *hand GS>Octβ1R* RNAi RU+ flies as well as trained RU- controls (ANOVA, p = 0.0021, p<0.001). **(K)** *hand GS Octβ1R* RNAi RU+ and RU- flies both have decreased pacing-induced cardiac failure after exercise training (Chi-squared, p = 0.0045, p = 0.0022). **(L)** Exercised trained *hand GS>Octβ1R* RNAi RU- flies increase fat body LysoTracker staining compared to unexercised siblings (ANOVA with Tukey multiple comparisons, p<0.0001), but RU+ have low fat body LysoTracker staining whether exercised or not.

RU- controls (Fig 2G–2I and 2L, S9D Fig). *Octβ1R* reduction in adult hearts did not affect adaptations in landing height or cardiac stress resistance, however (Fig 2J and 2K).

## Octopamine β2 and β3 receptor expression in muscle separately coordinate exercise adaptations

Adult-specific reduction of neither *Octβ2R* nor *Octβ3R* significantly affected baseline endurance (Fig 3A and 3G). In contrast, knockdown of either receptor prevented adaptation to chronic exercise (Fig 3B and 3H). RNAi against *Octβ2R* in adult muscle caused unusual climbing phenotypes, with unexercised knockdown flies climbing faster than controls, but actually becoming slower when exercise-trained (Fig 3C). Muscle-specific *Octβ3R* knockdown flies had normal baseline performance, but exercise training significantly worsened their climbing speed (Fig 3I). Knockdown of *Octβ2R* or *Octβ3R* in adult muscle also reduced normal increases in LysoTracker staining in trained RU+ flies, (Fig 3F and 3L S9E and S9F Fig) while exercised RU- flies adapted with exercise normally in all assessments (Fig 3, RU- EX). Muscle specific *Octβ2R* RNAi did not block adaptations to flight performance (Fig 3D), but *Octβ2R* was required in muscles for the cardioprotective effect of exercise (Fig 3E). In contrast, muscle–specific *Octβ3R* RNAi did not block exercise-induced cardiac improvements (Fig 3K), but did prevent improvements to flight (Fig 3J).

## Octopamine β3 receptor is important for fat body homeostasis

All 4 octopamine receptors tested here are known to be present in adult brain, with developmental and tissue-specific activities that are context dependent [17]. Among tissues tested here, *Octβ3R* transcript was only detected in adult fat body, and has been previously reported to be present at low levels in hindgut and Malpighian tubules [17]. *Octβ3R* RNAi in adult fat body had no effect on baseline or post-training endurance, exercise-induced climbing improvement, or resistance to pacing-induced cardiac stress (Fig 4A, 4B, 4C and 4E). The major effect of *Octβ3R* knockdown in adult fat body was a cell-autonomous block of Lyso-Tracker staining accumulation after exercise (Fig 4F and 4G). Perhaps surprisingly, *Octβ3R* expression was also required in adult fat body for exercise-dependent increases in flight performance (Fig 4D).

Given that exercised male flies increase lysosomal activity and mitochondrial turnover in the fat body [8,13] and flies with defects in fatty acid metabolism increase lipolysis following exercise [7], we tested whether flies with reduced *Octβ3R* expression in adult fat body also failed to upregulate autophagy by examining dAtg8-II/I ratio via Western blot. Autophagy downregulation is observed by a decrease in the dAtg8-II/I ratio, indicating a reduction in activated dAtg8-II, while restoration or upregulation is represented by an increase in the activated form [32]. We also tested exercise trained octopamine-fed flies, as both exercise and octopamine have been previously shown to increase autophagy and lipolysis in multiple flying insect

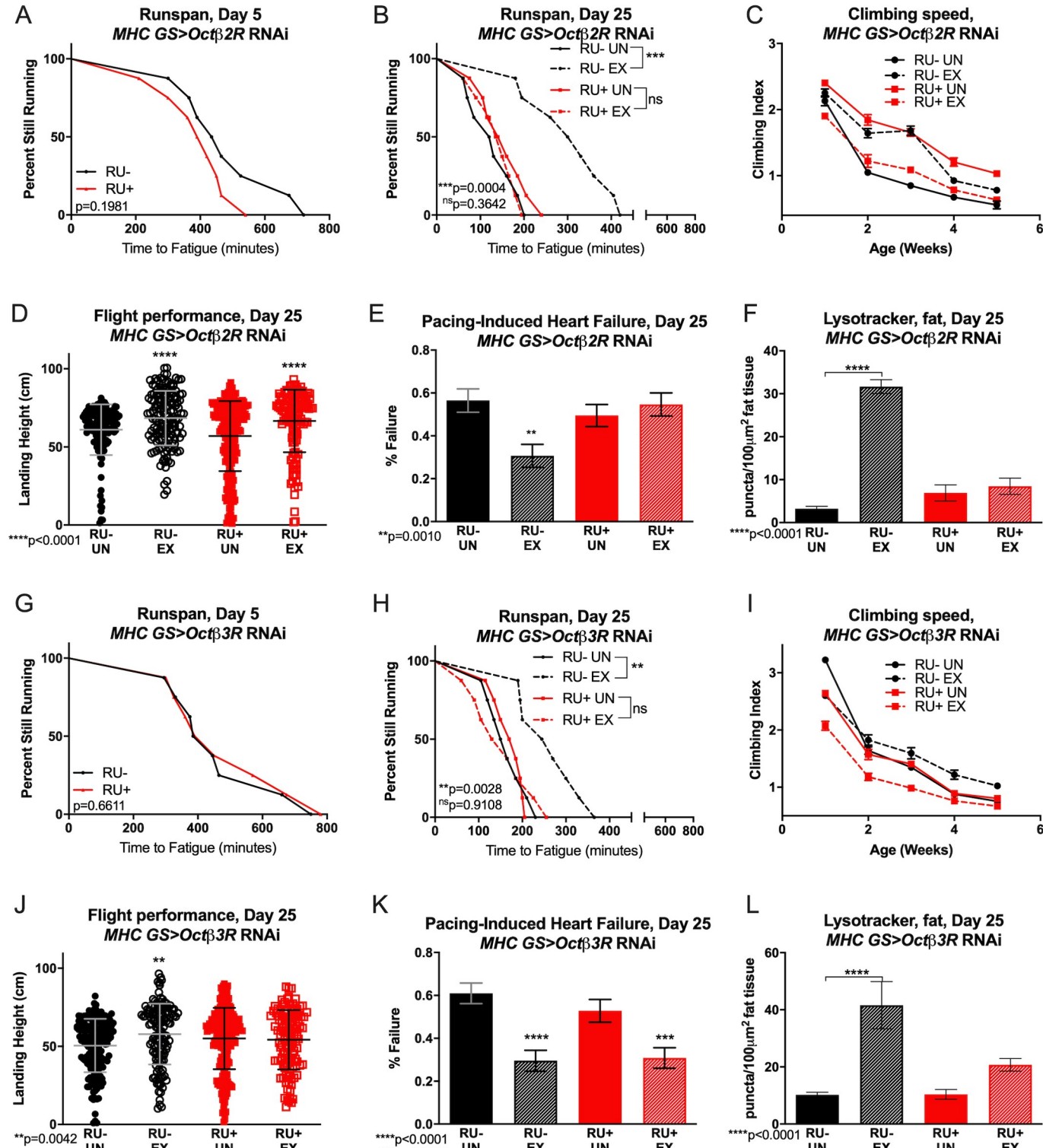

**Fig 3. Octopamine *β2* and *β3* receptors in muscle coordinate tissue-specific exercise adaptations in *Drosophila*. (A)** *MHC GS>Octβ2R* RNAi RU+ flies have similar endurance to uninduced RU- controls on day 5 of adulthood (log rank, p = 0.1981). **(B)** *MHC GS>Octβ2R* RNAi RU+ flies do not improve endurance after exercise training (log-rank, p = 0.3642) and resemble untrained flies (log-rank, p = 0.9571) while RU- exercised control flies retain greater endurance compared to unexercised, uninduced controls (log-rank, p = 0004). **(C)** *MHC GS>Octβ2R* RNAi RU- control flies show typical longitudinal climbing response during and after the exercise program is complete, with increased speed at weeks 2–4 (2-way ANOVA, exercise effect, p<0.0001). Exercise training in *MHC GS>Octβ2R* RNAi RU+ flies impairs normal increases in climbing speed, and climbing index is lower than untrained RU- flies at week 1 (2-way ANOVA,

genotype effect, p = 0.0297) and statistically similar at weeks 2 and 4 (2-way ANOVA, genotype effect, p = 0.1559, p = 0.5446). Untrained *MHC GS>Octβ2R* RNAi RU+ have statistically higher climbing index than trained RU+ siblings at all ages (2-way ANOVA, exercise effect, p<0.0001). **(D)** Both *MHC GS>Octβ2R* RNAi RU+ and RU- flies have improved landing height after exercise training compared to unexercised siblings (ANOVA with Tukey post-hoc comparison, p<0.0001). **(E)** Untrained *MHC GS>Octβ2R* RNAi RU+ flies have similar cardiac failure rate in response to external electrical pacing as untrained RU- controls (Chi-squared, p = 0.3478), but do not improve cardiac stress resistance after training (Chi-squared, p = 0.8948). Exercised *MHC GS>Octβ2R* RNAi RU- flies have lower failure rate than age-matched, unexercised RU- siblings (Chi-squared, p = 0.0010). **(F)** Untrained *MHC GS>Octβ2R* RNAi RU+ flies have low levels of LysoTracker staining in adult fat body, and exercise-trained siblings have statistically similar levels of fat-body LysoTracker staining (ANOVA with Tukey post-hoc, p = 0.9029). RU- control flies respond to exercise with increased fat-body Lysotracker staining in comparison to age-matched, unexercised siblings (p<0.0001). **(G)** *MHC GS>Octβ3R* RNAi RU+ flies have similar endurance to *MHC GS>Octβ3R* RNAi RU- control flies (log rank, p = 0.6611). **(H)** *MHC GS>Octβ3R* RNAi RU+ resemble untrained RU- controls whether exercised or not (log-rank, p = 0.9120) and do not improve with training (log-rank, p = 0.9108). RU- uninduced controls have higher endurance after exercise (log-rank, p = 0.0028). **(I)** *MHC GS>Octβ3R* RNAi RU- control flies respond to exercise with increased climbing speed in weeks 3–5 (2-way ANOVA, exercise effect, p≤0.0181) and *MHC GS>Octβ3R* RNAi RU+ untrained flies are only significantly slower in climbing speed in comparison to untrained RU- controls at week 1 (2-way ANOVA, genotype effect, p<0.0001). Exercise further reduces climbing performance in *MHC GS>Octβ3R* RNAi RU+ in weeks 1–3 to levels below RU+ and RU- untrained groups (2-way ANOVA, genotype effect, p<0.0001). **(J)** *MHC GS>Octβ3R* RNAi RU+ flies do not improve landing height after exercise and resemble untrained RU- controls (ANOVA with Tukey post-hoc comparison, p = 0.3484), while exercised RU- flies land higher than unexercised RU- siblings (ANOVA with Tukey post-hoc comparison, p = 0.0042). **(K)** Both *MHC GS Octβ3R* RU- and RU+ flies adapt to exercise training with reduced cardiac failure after external electrical pacing compared to unexercised controls (Chi-squared, p<0.0001, p = 0.0026), and *MHC GS Octβ3R* RU+ flies have similar failure rate to RU- controls in untrained conditions (Chi-squared, p = 0.9932). **(L)** *MHC GS>Octβ3R* RNAi RU+ show similarly low LysoTracker staining in adult fat body whether exercised or not, and RU- controls have increased fat-body lysosomal activity (ANOVA with Tukey post-hoc, p<0.0001).

species [20,33]. dAtg8-II/I ratios trend toward an increase in exercise-trained flies and were significantly higher in OA-fed whole flies, and this effect was blocked when *Octβ3R* was knocked down in adipose tissue (Fig 4H and 4I).

## OA feeding rescues phenotypes of some, but not all, octopamine receptor knockdowns

Either 5μM octopamine (OA) feeding or intermittent OA-ergic neuron activation is sufficient to replicate exercise adaptations in sedentary *Drosophila* [8]. To test whether surplus OA could overcome the effects of tissue-specific octopamine receptor depletion, we repeated the RNAi experiments above but fed OA to half the flies. We selected *MHC GS>Octβ2R* and *MHC GS>Octβ3R* RNAi flies for feeding tests since these lines were directly comparable (same driver) but blocked distinct and separable exercise adaptations. Runspan was tested on day 5 post-eclosion, after 3 days of feeding with 5μM OA or vehicle and/or RU486 if Gene-Switch Gal4 was employed. Drug/ vehicle feeding continued until the end of experimentation. A summary of OA-feeding+exercise results is in S1 Table. As in the first repetition, knockdown of *Octβ2R* in skeletal muscle prevented exercise-induced improvements. OA-fed control flies had performance characteristic of exercised flies, as previously observed [Figs 8] (S8B Fig). (Compare 5A to S1B RU-EX). However, OA-feeding did not restore the exercise response to flies lacking *Octβ2R* in muscle, as measured by endurance (compare Fig 5A RU+ OA-fed flies to S8B RU+) or climbing speed (Fig 5B, S8C Fig).

As above, improvements to flight performance after exercise did not require *Octβ2R* in adult muscles (S8D Fig, Fig 3D). Both exercise training and OA feeding were able to improve flight to levels that were similar to trained control flies (Fig 5C, compare to S8D Fig). Confirming results shown above, *Octβ2R* was required in muscle for cardio-protective effects of exercise, and for increased adipose lysosomal activity (S8E and S8F Fig) OA-feeding was completely unable to rescue these effects, although it successfully mimics exercise in OA-fed RU- controls (Fig 5D–5F). These results indicate that *Octβ2R* is absolutely required in muscle for chronic exercise to increase endurance, cardiac performance and lysosomal activity, even if exogenous OA is supplied.

## Octβ3R

Muscle-specific *Octβ3R* knockdown does not alter baseline endurance (S8G Fig). Following endurance training, *Octβ3R* knockdown flies again failed to increase endurance. However,

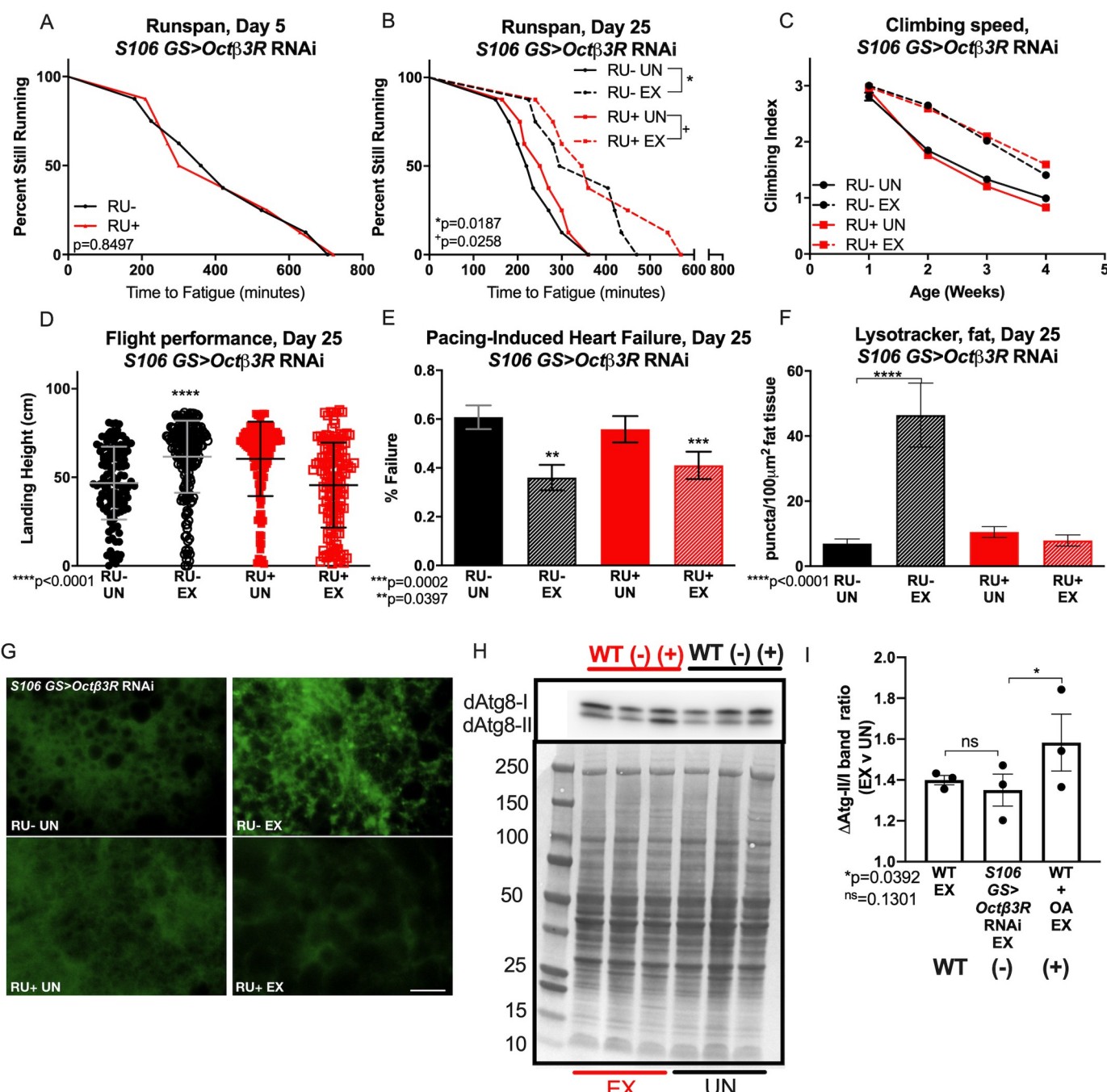

**Fig 4. Effects of reduced fat body Octopamine *β3* receptors on exercise adaptations are largely cell autonomous. (A)** *S106 GS>Oct β3R* RNAi RU+ flies have endurance that is similar to RU- control siblings (log-rank, p = 0.8497). **(B)** Both trained *S106 GS>Octβ3R* RNAi RU+ and trained uninduced RU- controls have better endurance after exercise training compared to unexercised cohorts (log-rank, p = 0.0258, p = 0.0187). **(C)** Both *S106GS>Octβ3R* RNAi RU- and RU+ flies respond to exercise with increased climbing speed in weeks 2–4 compared to untrained cohorts (2-way ANOVA, exercise effect, p<0.0001). **(D)** Exercise training does not improve flight performance in *S106 GS>Octβ3R* RNAi RU+ flies, which have similar landing height to untrained RU- controls (ANOVA with Tukey post hoc, p = 0.9756). *S106 GS>Octβ3R* RNAi RU- flies have increased landing height after exercise (p<0.0001). **(E)** Exercise-trained *S106 GS>Octβ3R* RNAi RU+ and RU- flies improve cardiac performance after pacing stress with lower failure rate (Chi-squared, p = 0.0002, p = 0.0397). **(F)** Exercised and unexercised *S106 GS>Octβ3R* RNAi RU+ flies have low levels of lysosomal activity in adult fat body, while RU- control flies have increased fat body LysoTracker staining in exercised groups in comparison to age-matched, unexercised siblings (ANOVA with Tukey post-hoc, p<0.0001). **(G)** Representative 40X confocal image of fat-body lysotracker staining. Scale bar = 20μm. **(H)** Whole exercise-trained flies of indicated strains were analyzed through immunoblotting of dAtg8 proteins. Direct blue was used for loading. Lanes for exercised groups are indicated in red, unexercised in black. Lane abbreviations: "WT": *w*[1118], "(-)":*S106 GS>Oct β3R* RNAi RU+, "(+)": *w*[1118] + OA feeding. **(I)** Relative levels of unprocessed (dAtg8-I) and processed (dAtg8-II) dAtg8 proteins were quantified by densitometry. The total volume of each band was quantified using ImageLab (Bio-Rad) and the amount of dAtg8-II was divided by the total signal from both dAtg8 bands in its respective lane. Bar graphs represent ratio metric change in 15kDa (dAtg8-II) band relative to matched, untrained cohorts (paired t-test, n = 3, means ± SEM).

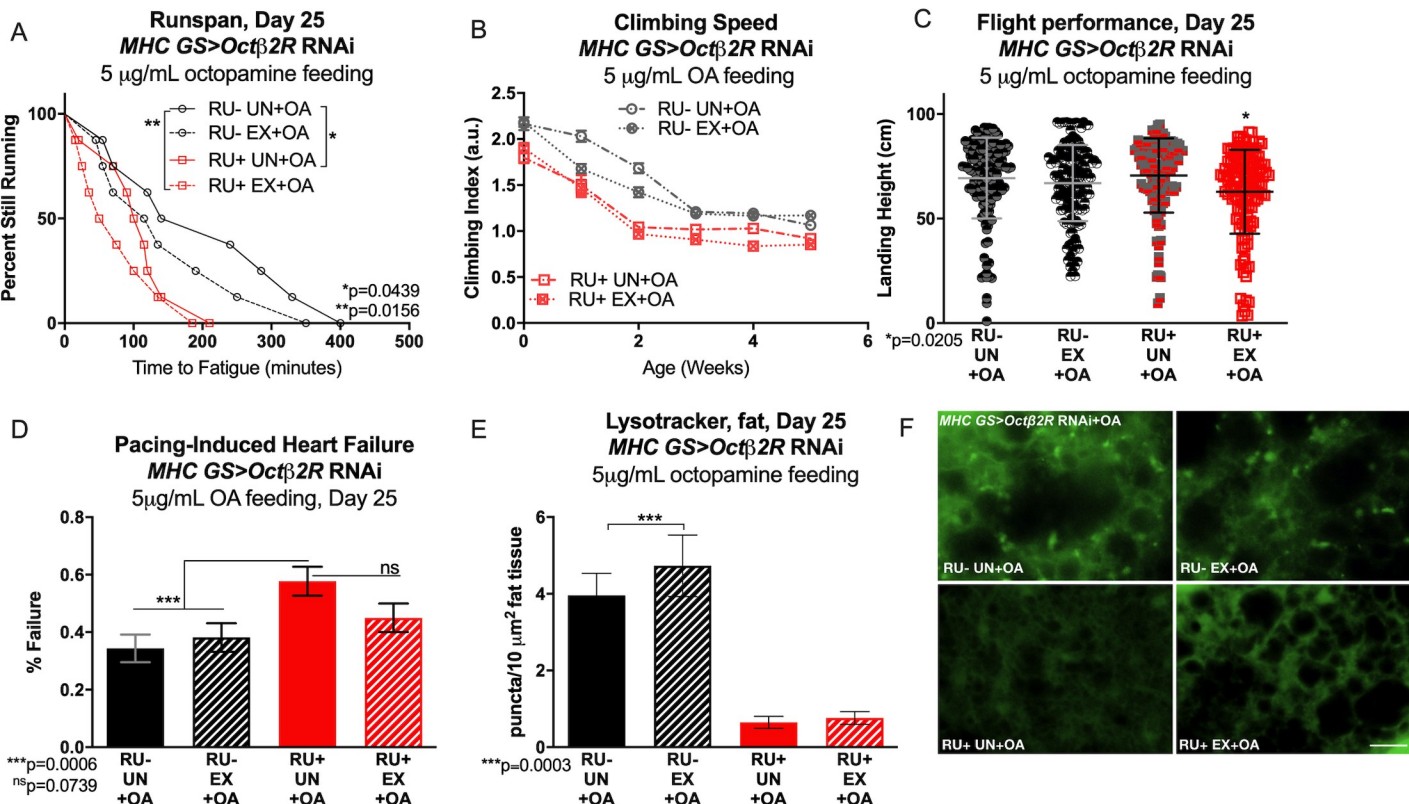

**Fig 5. Octopamine feeding rescues flight performance in *MHC GS>Octβ2R* RNAi flies.** (A) 5 μM OA-feeding in trained and untrained RU- *MHC GS>Octβ2R* RNAi flies increases endurance in comparison to RU+ flies (log-rank, p = 0.0298), with untrained, *MHC GS>Octβ2R* RNAi RU- flies receiving the most statistically significant benefit from OA-feeding (p = 0.0156, n = 8 vials of 20 flies for each cohort). (B) Similarly, OA feeding increases longitudinal climbing speed across ages in *MHC GS>Octβ2R* RNAi RU- untrained flies with the highest effectiveness (2-way ANOVA, genotype effect, p≤0.0023, n≥100 flies for each cohort, error bars = SEM), while RU+ do not increase climbing index with OA feeding or exercise. (C) In contrast, both OA-feeding and exercise improve landing height in *MHC GS>Octβ2R* RNAi RU- and RU+ flies, although exercise causes a slight but significant decrease in landing height in trained *MHC GS>Octβ2R* RNAi RU+, OA-fed flies (ANOVA with Tukey-post hoc, p = 0.0205, n≥139 flies, error bars = SD). (D) Exercise and OA-feeding confer lower failure rate in *MHC GS>Octβ2R* RNAi RU- flies when compared to *MHC GS>Octβ2R* RNAi RU+ siblings fed 5μM OA whether exercised or not (Chi-squared, p = 0.006, n≥97, error bars = SEM). (E) Lysosomal activity remains low in the fat body *MHC GS>Octβ2R* RNAi RU+ flies independent of exercise or OA-feeding, but is increased in the fat body of OA-fed or exercised, OA-fed *MHC GS>Octβ2R* RNAi RU- siblings (ANOVA with Tukey post-hoc, p≤0.0003, n = 10, error bars = SEM). (F) Representative 40X confocal image of fat-body lysotracker staining. Scale bar = 20μm.

they did increase performance when supplemented with OA, in both exercised and unexercised cohorts, suggesting that the requirement for *Octβ3R* can be partially circumvented by other receptors, if exogenous OA is present (Compare Fig 6A RU+ to S8H Fig RU+).

Exogenous OA was able to stimulate performance of muscle-specific *Octβ3R* RNAi flies, with both exercised and unexercised cohorts responding to OA feeding. OA feeding also improved heart performance in *MHC GS>Octβ3R* RNAi flies, with exercised and unexercised RU+ groups receiving as much cardioprotection as RU- controls (Fig 6D). By contrast, OA feeding did not alter the effect of muscle-specific *Octβ3R* knockdown on flight performance or lysosomal activity (Fig 6C, 6E and 6F; S8J–S8L Fig).

Thus, in general, exercise adaptations that were unaffected by a specific knockdown responded to OA feeding as normal, but adaptations that required a particular receptor were not rescued by OA feeding. This strongly suggests that some receptors are specifically required for particular exercise adaptations, and not all OA receptors are interchangeable in this context.

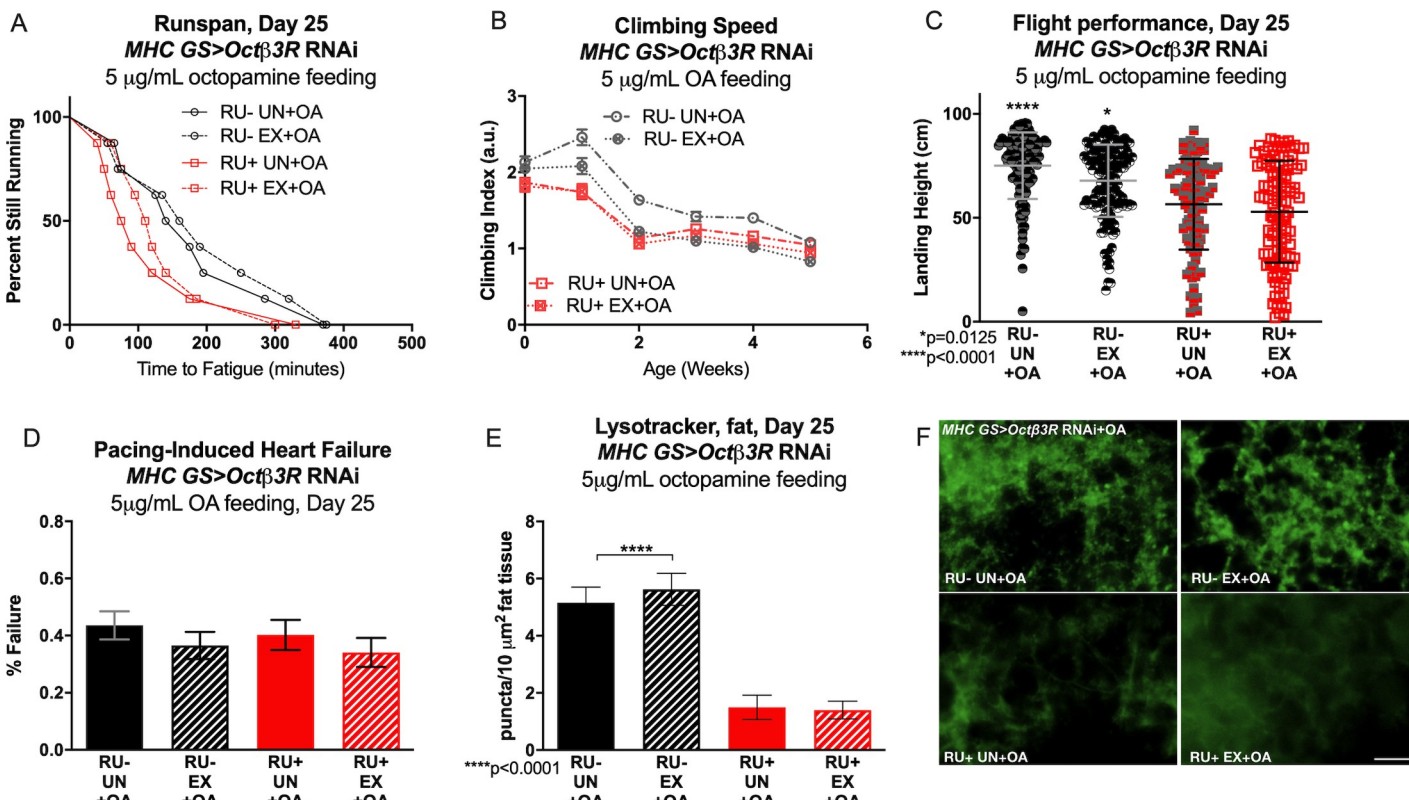

**Fig 6. Octopamine feeding rescues cardiac performance in *MHC GS>Octβ3R* RNAi flies. (A)** Endurance is improved in OA-fed *MHC GS>Octβ3R* RNAi RU- flies whether exercised or not, and RU+ OA-fed siblings have endurance that is not significantly different (log-rank, p = 0.3329, n = 8 vials of 20 flies for each cohort). **(B)** Climbing speed is similarly enhanced in OA-fed *MHC GS>Octβ3R* RNAi RU- EX and RU+ flies independent of training, although RU- UN flies fed 5μM OA receive significantly greater benefits across ages (2-way ANOVA, genotype effect, p≤0.0149, n≥100 flies for each cohort, error bars = SEM). **(C)** *MHC GS>Octβ3R* RNAi RU + flies have lower landing height than *MHC GS>Octβ3R* RNAi RU- flies independent of OA-feeding or exercise training plus OA-feeding (ANOVA with Tukey post-hoc, p<0.0001, p = 0.0125, n≥106, error bars = SD). **(D)** *MHC GS>Octβ3R* RNAi RU+ flies, however, have equally low failure rate to *MHC GS>Octβ3R* RNAi RU- flies that are exercise trained and/or OA-fed (Chi-squared, p = 0.6264, n≥87). **(E)** Trained and untrained OA-fed *MHC GS>Octβ3R* RNAi RU- flies have high fat-body LysoTracker staining, but LysoTracker staining is low in trained and untrained *MHC GS>Octβ3R* RNAi RU+ flies (ANOVA with Tukey post-hoc, p<0.0001, n = 10, error bars = SEM). **(F)** Representative 40X confocal image of fat-body lysotracker staining. Scale bar = 20μm. For endurance tests, n = 8 vials of ≥20 flies, all experiments performed in duplicate or triplicate. Runspan graphs indicate representative repetition. For climbing speed, n≥100 flies for all climbing experiments, performed in duplicate or triplicate. Error bars indicate SEM, climbing graphs indicate representative repetition. In acute flight performance assay, n≥121 flies unless otherwise indicated in legend. Experiments performed in duplicate or triplicate, error bars indicate SD, flight graphs indicate representative repetition. n≥71 for all external electrical pacing experiments. Each performed in duplicate or triplicate. Pacing graphs are representative repetitions, error bars = SEM. LysoTracker experiments performed with n of 5–10, in duplicate or triplicate. LysoTracker graphs are representative repetitions, error bars = SEM.

## Discussion

Octopamine signaling is a vital mediator of behavior and metabolism, and is critical for exercise adaptation in *Drosophila* [8,30]. OA directly affects muscle contractility in larval body wall muscle [34,35], metabolism [36], mobility in response to starvation [37] and fat storage [36], all of which may be important mechanisms modulating exercise adaptations. OA signals through various receptors that have been found to regulate essential processes from egg-laying to sleep, metabolism, learning and memory, and social aggression [38–42].

Octopamine is analogous to vertebrate norepinephrine, and noradrenergic signaling is known to be important in the human exercise response. While transient increases in OA-ergic signaling are sufficient to replicate exercise adaptation in sedentary *Drosophila* [8], prolonged effects of increased catecholaminergic signaling in humans would have adverse effects on blood pressure and heart rate [43,44]. Indeed, in our studies combining OA-feeding and

endurance exercise, we often see less benefit than OA-feeding or exercise alone, suggesting that activation of OA-receptors may become deleterious if activated at too high a level even in invertebrate systems. Here, we demonstrate that OA acts during exercise to stimulate autophagic flux in *Drosophila*. Taken together, our observations suggest that adrenergic signaling is an important mechanistic part of the conserved adaptive response to endurance exercise.

Although OA-receptors are thought to act through highly conserved canonical signaling pathways [45–47], we find strong evidence that their activity is not interchangeable in the context of exercise, as has been previously demonstrated in the context of female reproduction (41). Knockdown of any of the four receptors tested here (a recently discovered receptor that responds to both OA and serotonin was not examined here [48]) eliminates some portion of the response to chronic exercise, and, in most cases, the response cannot be rescued by stimulating other receptors with OA feeding. This is true even when knockdown takes place in tissues that express multiple receptors, strongly implying that downstream effects of these receptors are not identical.

Another key finding here is that several receptor knockdowns produced clear tissue-non-autonomous effects, particularly knockdowns in skeletal or cardiac muscle. These effects could result from improved muscle and heart performance that alters the overall metabolic environment during chronic exercise; for example, improved cardiac performance could improve circulation to other tissues. An intriguing alternative, albeit non-mutually exclusive hypothesis would be that OA-receptors promote release of circulating factors from muscle or heart that affect metabolism in the fat body. These hypotheses are currently under further investigation. These results are consistent with the prior observations that restoration of exercise adaptations to female flies required masculinization of all Tdc2-expressing neurons, no smaller subset was capable of this effect (8). This strongly implies that the effect of OA is not solely mediated by a particular circuit, but at least in part requires release of OA into circulation, where it can then be received by receptors in various tissues.

An unexpected finding was the requirement for *OAMB* in the heart, where it has not previously been reported to be expressed. Here, *OAMB* reduction in heart and muscle prevented exercise-dependent adaptations in endurance, climbing speed and heart performance but did not negatively affect flight or fat body lysosomal activity. OAMB has been implicated in reward signaling [49], but more recently has been directly linked to behavior and metabolism via insulin like signaling [50] and sugar overconsumption studies [51]. Taken together, those studies and ours suggest that *OAMB* is modulating exercise adaptations by responding to energy needs via cell- and non-cell autonomous mechanisms. In the myocardium, *OAMB* may also mediate increases in heart rate during a bout of exercise itself, as adrenergic signaling increases heart rate in multiple species, including humans [26], and OA can increase larval heart rate in *Drosophila* [52]. It is possible that increased stimulation of heart rate may have secondary effects in other tissues by changing the rate of circulation of nutrients, hormones, or OA itself.

We find that *Octβ2R* and *Octβ3R* are important in adult *Drosophila* muscle for exercise adaptations to endurance and climbing speed. Both muscle-specific knockdowns also have tissue non-autonomous effects on the fat body. When we combine OA-feeding with exercise, we see a partial rescue of both climbing and endurance in *Octβ3R* knockdowns, an improvement that is not seen during exercise alone. This suggests that, unlike the other receptors tested here, *Octβ3R* activity is partially redundant and can be supplemented by other receptors if ligand dose is high enough.

Adding further complexity to mapping the specific roles of each receptor, there are several cases where tissue-specific knockdowns produce effects that are not predicted by the results of ubiquitous knockdown. For example, we find that fat-specific knockdown of *Octβ3R* blocks increased Lysotracker activity during exercise, which is surprising because we previously

reported that ubiquitous knockdown of *Octβ3R* does not (8). Under wild-type conditions, *Octβ3R* is the primary OA receptor in adipose tissue. It could be that disruption of *Octβ3R* in both muscle and fat induces compensatory effects to maintain lysosomal activity in the absence of *Octβ3R*. Alternately, it is possible that the highly efficient and consistent knockdown provided by the ubiquitous *Tub5-Gal4* induces a compensatory response that is not induced by the gradually accumulating knockdown driven by *S106-Gal4* (S1 Fig). This phenomenon was not limited to *Octβ3R* knockdown, as muscle-specific *Octβ1R* knockdown unexpectedly reduced endurance more than ubiquitous *Octβ1R* knockdown. Whereas the role of OA in stimulating lipolysis is thought to involve signaling through PKA/cAMP, OA also regulates muscle contractility through its effects on $Ca^{2+}$/IP3/CaMK signaling [53]. It may be that knockdowns of different strength or in different tissue combinations affect signaling through these pathways differentially. Alternately, tissue-non-autonomous effects of OA receptors may contribute to differences in these phenotypes in complex ways. Further investigation of downstream factors activated through OA-ergic signaling in different tissues during exercise and how those change during various manipulations will be necessary to resolve these questions unambiguously.

 *Octβ1R* reduction in muscle or heart produced the most broadly deleterious phenotypes of all, reducing endurance as early as day 5, and negatively affecting all parameters of exercise adaptation. It is worth noting that *Octβ1R* transcript was previously found to be upregulated in both endurance-exercised and longevity selected flies, indicating its importance in the preservation of healthy physiology [5].

 We have successfully mapped several specific adrenergic receptor requirements for endurance exercise adaptations in *Drosophila* (Fig 7). Further understanding of tissue-specific requirements for adrenergic signaling moves us closer to comprehensive mechanisms that govern exercise responses and potentially contribute to genetic differences in individual exercise responses.

## Materials and methods

### Fly stocks and maintenance

All fly lines were reared and aged at 25˚C; 50% humidity with a 12-hour light-dark cycle and provided with a standard 10% yeast/10% sucrose diet unless otherwise indicated. All RNAi lines were validated in combination with each driver before and after exercise. All *Drosophila* lines were from the Bloomington *Drosophila* Stock Center or Vienna *Drosophila* RNAi Center with the following exceptions: *hand GS Gal4* and *MHC GS* were obtained from Rolf Bodmer (Sanford Burnham Medical Research Institute) and *S106 GS* was obtained from Marc Tatar (Brown University). BDSC lines were *w1118* (BDSC3605) and *OAMB RNAi* (BS31171). VDRC lines were *Octβ1R* RNAi (v47895), *Octβ2R* RNAi (v8486) and Oct*β3R* RNAi (v101189). All driver lines have been previously characterized [54–56]. All experiments were performed using Gene-switch Gal4. For Gene-switch experiments, genetic background effects were controlled for by using RU- flies of the same background as the negative control. Raw data from all experiments throughout the manuscript is provided in S10 Fig.

### Drug treatment

For gene-switch experiments, adult progeny were age-matched by collecting within 2 hours of eclosion over a 72 hour time period and immediately transferred into vials containing 5mL standard medium. Populations were split into control RU- and experimental RU+ groups on the 2nd day and transferred into vials containing 5mL medium containing either 70% ethanol vehicle or 100 μM mifepristone (RU486) (Cayman Chemical, Ann Arbor, MI), respectively.

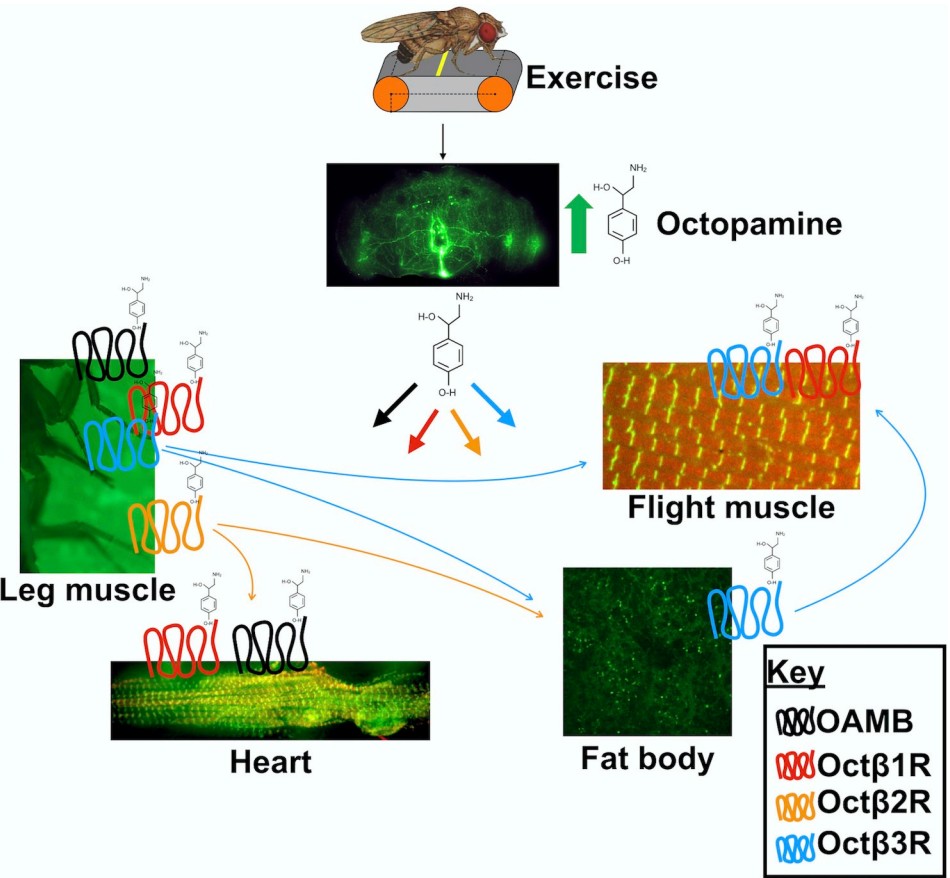

**Fig 7. Summary of tissue autonomous and non-autonomous effects of octopamine receptor activation during endurance exercise.** Endurance exercise in *Drosophila* increases octopaminergic signaling (thick arrows), activating receptors (color coded, see key) in target tissues which are required for beneficial adaptations. Cell non-autonomous effects proposed in study are indicated by thin arrows in colors corresponding to activated receptor. Lab-generated images clockwise from top, Brain: *Tdc2-GFP*, max projection (20X), IFM: *kettin-GFP*, Alexa-fluor 494 phalloidin (100X), Fat body: LysoTracker, *w^1118* EX (40X), Heart: *kettin-GFP*, Alexa-fluor 494 phalloidin (20X), Legs: actin-GFP (6X).

Experimental and control flies were then housed at 25ºC on either RU486 or vehicle until experimentation.

Flies fed octopamine were treated similarly to gene-switch experiments but were collected within a single 24-hour window immediately after eclosion and housed on SY10 food containing 5 µg/mL octopamine (Sigma-Aldrich, St. Louis, MO), or an equal volume of ddH$_2$O vehicle.

## Exercise training

Cohorts of at least 800 flies were collected under light CO$_2$ anesthesia within 2 hours of eclosion and separated into vials of 20. Flies were then further separated into 2 large cohorts of at least 400 flies divided into exercised and unexercised groups. If OA-feeding was employed, cohort size was doubled, and flies were divided into 4 cohorts: OA/UN, OA/EX, Vehicle/UN, Vehicle/EX. The unexercised groups were placed on the exercise training device but were prevented from running by the placement of a foam stopper low in the vial. The stopper is returned to the top of the vial at the conclusion of daily training. The exercise device drops the

vials of flies every 15 seconds, inducing a repetitive negative geotaxis response. Exercised flies are free to run to the top of the vial. Daily time of exercise followed the previously described ramped program [11].

All exercised and unexercised cohorts were assessed for speed, endurance, cardiac performance, flight, and fat body Lysotracker.

### Endurance

Climbing endurance was measured using the fatigue assay described previously [9]. Eight or sixteen vials of flies from each cohort were subjected to the fatigue assay at two time points: once on day 5 and once on day 25 of adulthood. For each assessment, the flies were placed on the Power Tower exercise machine and made to climb until they were fatigued. Monitored at 15 min intervals, a vial of flies was visually determined to be fatigued when 20% or fewer flies could climb higher than 1 cm after four consecutive drops. A minimum of 8 vials containing 20 flies each was used for each fatigue assessment with each vial plotted as a single datum. Each experiment was performed in duplicate or triplicate, and runspans were scored blindly when possible. The time from the start of the assay to the time of fatigue was recorded for each vial, and the data analyzed using log-rank analysis in GraphPad Prism (San Diego, CA, USA).

### Climbing speed

Adult flies were collected with light $CO_2$ anesthesia within 2 hours of eclosion and housed in appropriate fresh food vials. Negative geotaxis was assessed in Rapid Negative Geotaxis (RING) assays in groups of 100 flies as described [9]. Flies were transferred to individual polypropylene vials in a RING apparatus and allowed to equilibrate for 1 minute. Negative geotaxis was elicited by sharply rapping the RING apparatus four times in rapid succession. The positions of the flies were captured in digital images taken 2s after stimulus. Images were analyzed using ImageJ (Bethesda, MD). The distance climbed by each fly was converted into quadrants using Microsoft Excel. The performance of each vial of 20 flies was calculated as the average of four consecutive trials to generate a single datum. Flies were longitudinally tested 5 times per week for 4–5 weeks to assess decline in negative geotaxis speed with age. Data were further consolidated into weekly performance. Between assessments, flies were returned to food vials and housed until the following RING test. Negative geotaxis results were analyzed using two-way ANOVA analysis with *post hoc* Tukey multiple comparison tests in GraphPad Prism (San Diego, CA, USA). All negative geotaxis experiments were performed in duplicate or triplicate.

### Flight performance

Flight was analyzed as in Sujkowski *et al*. 2017 [8]. Duplicate or triplicate cohorts of at least 50 flies were exercise trained in narrow vials housing groups of 20 age-matched siblings. Acrylic sheeting with paintable adhesive was placed in the flight tube, and fly cohorts were ejected into the apparatus to record flight performance and subsequent landing height after release. Fly cohorts were introduced to the flight tester one vial at a time using a gravity-dependent drop tube in order to reduce variability. After a full cohort of flies was captured on the adhesive, the sheeting was removed to a white surface in order to digitally record the landing height of each fly. Flies with damaged wings were censored from final analysis to control for mechanical stress not related to training performance. Images were analyzed using ImageJ. Landing height was averaged and compared in Prism using ANOVA with Tukey post-hoc comparison.

## Cardiac pacing

25-day old flies were removed from appropriate experimental cohorts and subjected to electrical pacing as in Wessells *et al.* [57]. The percentage of fly hearts that responded to pacing with either fibrillation or arrest were recorded as "% failure". Pacing-induced failure rate is a marker for stress sensitivity and characteristically declines with age [31,58]. Endurance exercise reduces cardiac failure rate across ages in trained male *Drosophila* [7,11,31]. Failing hearts are scored as "1" and hearts that respond to pacing stress with normal beating are scored as "0". Averages are analyzed by Chi-squared test for binary variables.

## Lysotracker

Lysotracker staining of adult fat bodies was performed as in Sujkowski *et al.* [8]. Adult flies separated by age, genotype, and or treatment were dissected, ventral side up, in room temperature PBS. Having exposed fat bodies, partially dissected flies were rinsed 1X in fresh PBS. Lysotracker green (Molecular Probes, Eugene, OR) was diluted to 0.01μM in PBS and applied to dissected preps for 30 seconds. Samples were washed 3 times in fresh PBS. Stained fat bodies were subsequently removed and mounted in Vectashield reagent (Vector Laboratories, Burlingame, CA, USA). Confocal imaging was done in the Department of Physiology Confocal Microscopy Core at Wayne State School of Medicine on a Leica DMI 6000 with a Crest X-light spinning disc confocal using a 63X oil immersion objective or widefield fluorescent 40X objective. Images were analyzed using ImageJ. A minimum of 10 samples were analyzed for each sample and duplicate or triplicate biological cohorts were assessed for each group. Data were subjected to ANOVA with Tukey post-hoc.

## Western Blotting

Triplicate biological cohorts of 3 whole flies per genotype/treatment were homogenized in boiling lysis buffer (50 mM Tris pH 6.8, 2% SDS, 10% glycerol, 100 mM dithiothreitol), sonicated for 15 seconds, boiled for 10 min, and centrifuged at 13,300 × g at room temperature for 10 min. Samples were electrophoresed on 4–20% gradient gels (Bio-Rad). Western blots were developed using the ChemiDoc system (Bio-Rad). Direct blue staining was used for total protein loading: PVDF membranes were submerged for 5 min in 0.008% Direct Blue 71 (Sigma-Aldrich) in 40% ethanol and 10% acetic acid. PVDF membranes were then rinsed briefly in 40% ethanol and 10% acetic acid solvent, then ultrapure water, air dried, and imaged using the ChemiDoc system. Anti-dAtg8a antibody (ab109364) was from obtained from Abcam. Blots were quantified using ImageLab software (Bio-Rad).

## qRT PCR

RNAi efficacy was confirmed tissue-specifically pre- and post-exercise for all Gal4-UAS RNAi combinations tested (S1A–S1C Fig). To control for non-specific effects of RNAi, physiological assessments for Gal4-UAS RNAi combinations in absence of receptor expression are included as S2–S7 Figs. cDNA was prepared using a Cells to CT Kit (Invitrogen) from 20 adult fly hearts, or indirect flight muscle (IFM) or fat body from 5 adult flies. Two independent cDNA extractions were prepared for each sample. Differences between genotypes were assessed by ANOVA. Primer sequences are listed below.

  5' OAMB- CGGTTAACGCCAGCAAGTG
  3' OAMB- AAGCTGCACGAAATAGCTGC
  5'Octβ1R GGCAACGAGTAACGGTTTGG
  3' Octβ1R TCATGGTAATGGTCACGGGC

5'Octβ2R TTAGTGTGCAAGTAACTGGGC
3' Octβ2R TGAGAAGTAGACATCGAGGCTG
5'Octβ3R TGTGGTCAACAAGGCCTACG
3' Octβ3R GTGTTCGGCGCTGTTAAGGA
5' act5C GGCGCAGAGCAAGCGTGGTA
3' act5C GGGTGCCACACGCAGCTCAT

Relative message abundance was determined by amplification and staining with SYBR Green I using an ABI 7300 Real Time PCR System (Applied Biosystems). Expression of *Actin5c* and corresponding RU- control flies were used for normalization.

## Supporting information

**S1 Fig. Confirmation of RNAi efficacy in target tissues.** Bars represent triplicate samples consisting of **(A)** 20 hearts, or samples from 5 flies consisting of **(B)** IFM or **(C)** adult fat body. Samples were assigned to exercised and unexercised groups and collected at 72 hours, prior to the first endurance test, and 25 days, after the conclusion of exercise training. qRT-PCR was performed after cDNA isolation from aforementioned tissues. Relative expression is calculated as ΔΔCT and analyzed using ANOVA with Tukey post-hoc. Gene expression is expressed in relation to uninduced RU- controls. See methods for primer sequences, isolation, purification and reaction conditions.
(TIFF)

**S2 Fig. Baseline endurance is unaffected by non-specific RNAi effects.** Neither *Octβ2R* nor *Octβ3R* were detected in adult *Drosophila* heart **(A, B)**, and *OAMB*, *Octβ1R* and *Octβ2R* **(C-E)** transcripts were not detectable in adult fat body. Day 5 endurance in RU+ flies of each of the aforementioned genotypes were not statistically different from their RU- control flies of the same age.
(TIFF)

**S3 Fig. Post-training endurance is unaffected by non-specific RNAi effects.** *hand GS>Octβ2R* RNAi and *hand GS>Octβ3R* RNAi RU+ flies respond to exercise with improved endurance **(A, B)** as do *S106 GS>OAMB* RNAi, *S106 GS>Octβ1R* RNAi *and S106 GS>Octβ2R* RNAi flies **(C-E)**. (log-rank, p-values indicated in panels).
(TIFF)

**S4 Fig. Negative RNAi controls adapt to exercise with increases in climbing speed.** Both RU+ flies and uninduced RU- controls respond to exercise training with faster climbing speed across ages in **(A)** *hand GS>Octβ2R* RNAi, **(B)** *hand GS>Octβ3R* RNAi **(C)** *S106 GS>OAMB* RNAi, **(D)** *S106 GS>Octβ1R* RNAi and **(E)** *S106 GS>Octβ2R* RNAi groups. (2-way ANOVA, exercise effect, p<0.0001 after week 2, all groups).
(TIFF)

**S5 Fig. Flight performance is increased in exercise-trained RNAi negative control flies.** Landing height is higher in exercise trained RU- and RU+ **A)** *hand GS>Octβ2R* RNAi, **(B)** *hand GS>Octβ3R* RNAi **(C)** *S106 GS>OAMB* RNAi, **(D)** *S106 GS>Octβ1R* RNAi and **(E)** *S106 GS>Octβ2R* RNAi flies. (ANOVA with Tukey multiple comparisons, p values indicated in panels).
(TIFF)

**S6 Fig. No non-specific RNAi effects on post-training adaptations to cardiac stress resistance.** *hand GS>Octβ2R* RNAi and *hand GS>Octβ3R* RNAi RU+ flies respond to exercise with improved tolerance to external cardiac pacing **(A, B)** as do *S106 GS>OAMB* RNAi, *S106*

*GS>Octβ1R* RNAi *and S106 GS>Octβ2R* RNAi flies **(C-E)**. (Chi-squared, p values indicated in panels).
(TIFF)

**S7 Fig. Fat body LysoTracker staining is increased exercise-trained RNAi negative control flies.** LysoTracker staining is higher in exercise trained RU- and RU+ **A)** *hand GS>Octβ2R* RNAi, **(B)** *hand GS>Octβ3R* RNAi **(C)** *S106 GS>OAMB* RNAi, **(D)** *S106 GS>Octβ1R* RNAi and **(E)** *S106 GS>Octβ2R* RNAi flies. (ANOVA with Tukey multiple comparisons, p values indicated in panels).
(TIFF)

**S8 Fig. Vehicle-fed *MHC GS>OctR* RNAi flies have reductions in endurance, speed, cardiac stress resistance and fat body LysoTracker staining. (A)** *MHC GS>Octβ2R* RNAi RU- and RU+ flies fed 5μM OA or vehicle for 72 hours have equivalent endurance at day 5-post eclosion (log-rank, p = 0.2790, n = 16 vials of 20 flies for each cohort). **(B)** OA-fed *MHC GS>Octβ2R* RNAi RU+ flies have endurance similar to untrained, vehicle-fed RU- flies whether exercised or not (log-rank, p≥0.2558). Uninduced, vehicle-fed exercised controls retain better endurance than unexercised siblings (log-rank, p = 0.0439, n = 8 vials of 20 flies for all cohorts). **(C)** Exercise-trained, vehicle-fed *MHC GS>Octβ2R* RNAi RU- flies have faster climbing than unexercised, vehicle-fed siblings across ages (2-way ANOVA, exercise effect, p<0.0001). Both exercised and unexercised vehicle-fed *MHC GS>Octβ2R* RNAi have reduced climbing speed in comparison to RU- groups up to the second week of training (2-way ANOVA, genotype effect, p<0.0001) and do not improve with training or vehicle feeding, having similar climbing speed to untrained, RU- vehicle-fed groups in later weeks (n≥100 for all cohorts, error bars = SEM). **(D)** Vehicle feeding does not affect adaptation to flight performance after exercise in either *MHC GS>Octβ2R* RNAi RU- or *MHC GS>Octβ2R* RNAi RU + flies, as both increase landing height in comparison to unexercised siblings (ANOVA with Tukey post-hoc, p<0.0001, n≥119, error bars = SD). **(E)** Cardiac failure rate in response to external electrical pacing is lower in exercise-trained, vehicle-fed *MHC GS>Octβ2R* RNAi RU- flies compared to age-matched, untrained siblings (Chi-squared, p = 0.0396). Vehicle-fed *MHC GS>Octβ2R* RNAi RU+ flies do not improve cardiac stress response after training (Chi-squared, p = 0.5367, n≥95, error bars = SEM). **(F)** Lysosomal activity remains similar to untrained siblings in the fat body of vehicle-fed, exercise-trained *MHC GS>Octβ2R* RNAi RU + flies, but is increased in vehicle-fed, exercised RU- flies (ANOVA with Tukey post-hoc, p≤0.0304, n = 10, error bars = SEM). **(G)** *MHC GS>Octβ3R* RNAi RU- and RU+ flies fed 5μM OA or vehicle for 72 hours have equivalent endurance at day 5-post eclosion (log-rank, p = 0.9092, n = 16 vials of 20 flies for each cohort). **(H)** OA-fed *MHC GS>Octβ3R* RNAi RU + flies have endurance similar to untrained, vehicle-fed RU- flies whether exercised or not (log-rank, p≥0.2204). Uninduced, vehicle-fed exercised controls retain better endurance than unexercised siblings (log-rank, p = 0.0401, n = 8 vials of 20 flies for all cohorts). **(I)** Exercise-trained, vehicle-fed *MHC GS>Octβ3R* RNAi RU- flies have faster climbing than unexercised, vehicle-fed siblings across ages (2-way ANOVA, exercise effect, p<0.0001). Both exercised and unexercised vehicle-fed *MHC GS>Octβ3R* RNAi have reduced climbing speed in comparison to RU- groups in the first week of training (2-way ANOVA, genotype effect, p<0.0001) and do not improve with training or vehicle feeding, having similar or worse climbing speed than untrained, RU- vehicle-fed groups in later weeks (n≥100 for all cohorts, error bars = SEM). **(J)** Vehicle feeding does not affect adaptation to flight performance after exercise in *MHC GS>Octβ3R* RNAi RU- flies, but *MHC GS>Octβ3R* RNAi RU+ flies have flight performance similar to vehicle-fed, untrained RU- flies whether exercised or not (ANOVA with Tukey post-hoc, p = 0.0001,, p≥0.2216), n≥117, error bars = SD). **(K)** Exercise adaptations to cardiac

stress resistance in response to external electrical pacing are not affected by vehicle feeding in *MHC GS>Octβ3R* RNAi RU- or RU+ flies compared to age-matched, untrained siblings, and both trained groups have lower failure rates after training (Chi-squared, p = 0.0373, RU- EX, p = 0.0003, RU+EX, n≥100, error bars = SEM). **(L)** Lysosomal activity remains similar to untrained siblings in the fat body of vehicle-fed, exercise-trained *MHC GS>Octβ3R* RNAi RU + flies, but is increased in vehicle-fed, exercised RU- flies (ANOVA with Tukey post-hoc, p≤0.0001, n = 10, error bars = SEM).
(TIFF)

**S9 Fig. Representative 40X Fat Body Lysotracker Images** Accompanying confocal images for lysotracker quantifications in main Figs 1–3. Scale bars = 20μm.
(TIFF)

**S10 Fig. Raw Data file.** Raw data is presented as a spreadsheet with one page for each main figure and supplemental figure. Within each page, experimental data is labelled by subheadings for assessment and genotype.
(XLSX)

**S1 Table. Summary Statistics of combinatorial treatment of 5μM OA feeding plus exercise training in selected RNAi lines.**
(DOCX)

## Acknowledgments

We acknowledge the Bloomington Stock Center for providing fly lines and FlyBase for sequence and genome information.

## Author Contributions

**Conceptualization:** Alyson Sujkowski, Robert Wessells.

**Data curation:** Alyson Sujkowski, Anna Gretzinger, Nicolette Soave.

**Formal analysis:** Alyson Sujkowski, Sokol V. Todi, Robert Wessells.

**Funding acquisition:** Sokol V. Todi, Robert Wessells.

**Investigation:** Alyson Sujkowski, Anna Gretzinger, Nicolette Soave.

**Methodology:** Alyson Sujkowski, Anna Gretzinger, Sokol V. Todi, Robert Wessells.

**Resources:** Sokol V. Todi.

**Writing – original draft:** Alyson Sujkowski.

**Writing – review & editing:** Alyson Sujkowski, Sokol V. Todi, Robert Wessells.

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
