## [Decision Letter · Decision Letter 0]

16 Sep 2019

Dear Dr Wessells,

Thank you very much for submitting your Research Article entitled 'Alpha- and beta-adrenergic octopamine receptors in muscle and heart are required for Drosophila exercise adaptations' to PLOS Genetics. Your manuscript was fully evaluated at the editorial level and by independent peer reviewers. The reviewers appreciated the attention to an important problem, but raised some substantial concerns about the current manuscript. Based on the reviews, we will not be able to accept this version of the manuscript, but we would be willing to review again a much-revised version. We anticipate that the revision will require additional experimentation (see below). We cannot, of course, promise publication at that time.

Please carefully and thoroughly address all reviewer comments, especially the serious concerns listed below.

Description and verification of the Gal4 lines (Reviewer #1). Discuss the mechanistic insights on how octopamine receptors regulate exercise response (Reviewer #1 & #3). Provide rationale for each experiment (Reviewer #2). Tissue-specificity of the Gal4 lines (Reviewer #2). Verification of RNAi knockdown efficacy (Reviewer #2 & #3). Several conclusions need to be revised to match to the results (Reviewer #2). Use other autophagy markers to verify lysotracker results (Reviewer #3).

If you decide to revise the manuscript for further consideration at PLOS Genetics, please aim to resubmit within the next 60 days, unless it will take extra time to address the concerns of the reviewers, in which case we would appreciate an expected resubmission date by email to plosgenetics@plos.org.

[LINK]

We are sorry that we cannot be more positive about your manuscript at this stage. Please do not hesitate to contact us if you have any concerns or questions.

Yours sincerely,

Hua Bai, Ph.D.

Guest Editor

PLOS Genetics

Gregory P. Copenhaver

Editor-in-Chief

PLOS Genetics

Reviewer's Responses to Questions

**Comments to the Authors:**

Reviewer #1: In this manuscript, Sujkowski et al. build upon previous work demonstrating OA is necessary for exercise adaptations following endurance training in Drosophila by identifying which octopamine receptors are required for these adaptations. They found that OAb2R is required in adult muscle while OAMB is necessary in the heart. The authors demonstrate adrenergic signaling is key to the adaptive response to endurance exercise which is a finding that has potential to be significant and applicable in other systems and in individuals who are at risk for inactivity-related disorders. The large datasets for every paradigm is impressive and the behavioral data is rigorous and well done. The figures are constructed in a creative and easy-to-interpret manner considering the amount of data and the fairly cumbersome genotypes for each graph. In addition, the use of the Gene Switch system alleviates potential developmental issues related to reductions in OARs.

My enthusiasm is dampened by the lack of images and description of the Gal4 lines as the validity of the results weighs heavily on the tissue-restricted expression of these lines. Secondly, I realize there is an immense amount of time invested in the clear and solid behavioral experiments, however the manuscript stops short of providing a mechanism to explain how individual OARs govern exercise response. This is likely underway or will be incorporated into the next publication but it is a weakness in this manuscript.

Corrections and comments:

1. The verification of OAR transcript expression should be described earlier as these results are critical. I found myself often wondering if an individual receptor is even found in the fat body for example.

2. A short sentence or two describing Runspan, etc., should be provided at the start of the relevant sections so a reader does not have to look up the reference to understand the assay.

2. A fifth receptor that is activated by OA has been described - https://www.ncbi.nlm.nih.gov/pubmed/28942992

3. Are the Gal4 lines thoroughly characterized and do not drive expression in any other tissue? In the Materials and Methods, the hand-Gal4 and MHC-GS-Gal4 lines are listed without references and as gifts. Even if the lines are characterized, it would be useful to show the expression patterns by crossing the Gal4 lines to stinger-GFP or His2A-GFP.

4. In the OAb2R negative geotaxis assays on page 5, it looks like the climbing phenotypes of the experimental males are not statistically different than controls. If this is correct than the sentence (line 103) should be rewritten. Currently it says the experimental flies climb faster than controls.

5. At the end of page 7, a summary sentence describing the results should be provided.

6. The UAS-RNAi lines and BDSC numbers are not provided.

Typos, etc.

1. page 4, line 92. Figure instead of Figures.

2. Statistics should be added to Figure 3G and the spacing fixed.

3. There is only one * in Fig. 4D but the p-value indicates p<0.0001 as in Fig. 4A.

4. Fig. 5F has Octb1R

5. line 160, pg. 7 there are words missing in the sentence.

Reviewer #2: The manuscript by Sujkowski et al. is a follow-up of their work published in Cell Report 2017 and describes the roles of each of four known octopamine receptors in three different tissues (skeletal muscle, heart and adipocytes) in various exercise responses. There are several interesting findings such as the effects of combined OA treatment and exercise training. However, the work in the manuscript seems largely incomplete, the experiments are overall not rigorous and lack rationale (for example, there is no rationale on why the three tissues were examined), and conclusions are not supported by the presented data, offering limited significance and impact. Specific points are as follows.

Major points:

1. The authors used three GAL4 lines – MHC-, hand- and S106-GAL4 for skeletal muscle, heart muscle and fat body, respectively. However, their expression patterns are not specific - for example, MHC-GAL4 is expressed in all muscle types including skeletal and cardiac muscles and Hand-GAL4 in all cardiac cells including muscle and non-muscle cells, and S106-GAL4 in the digestive system as well. Additional GAL4 lines with more restricted expression patterns would be needed to make solid conclusions on the tissue-specific roles of each receptor.

2. RNAi efficacy is not documented. The authors note that each RNAi line was tested in the previous study. However, the previous study was done using da-GAL4, which is a strong driver (i.e. high expression level and ubiquitous). GAL drivers have different expression levels in different tissues, and more importantly RNAi efficacy varies depending on a tissue type. Thus, each driver on each receptor knockdown, particularly temporal knockdowns induced by 72 hrs RU486 feeding, would need to be confirmed by q-RT-PCR or in situ hybridization in the tissue type under study.

3. There are genetic mutants available for each octopamine receptor. It would be of great importance to examine the genetic mutants and tissue-specific rescues to verify the noted roles of each receptor in a particular tissue.

4. In all result sections/figures, the following receptor knockdowns/GAL4 driver are missing.

MHC-GS-GAL4: Octb1R alone

Hand-GAL4: Octb2R, Octb3R alone

S106: Octb1R, Octb2R and OAMB

4.1. When Octb3R/1R knockdown causes a phenotype, the conclusion cannot be made as to whether Octb3R or Octb1R alone, or combined actions of Octb1r and Octb3r are important in the absence of individual Octb1R and Octb3R knockdowns.

4.2. There is no rationale for knocking down Octb1R and Octb3R together. If the goal is to examine the effects of multiple receptor knockdowns, other receptor combinations are missing.

4. 3. For MHC and S106, GS-GAL4 is used for adult stage-specific knockdown. While Hand-GS-GAL4 is available, Hand-GAL4 is used in the study and no rationale is provided for it.

5. In the study on adaptation to chronic exercise, the duration of RU486 treatment is not noted. Regardless whether RU486 was treated for 3 days or throughout chronic exercise training (25 days), the endurance test on “acute” exercise should be done on Day 25 with the same RU486 treatment regime since the levels of octopamine receptor knockdowns are likely different on Day 5 (acute) and Day 25 (chronic). Otherwise the role of the receptors on acute and chronic exercise cannot be compared.

6. The experiments on the combined OA treatment and chronic exercise training are interesting. However, there is no rationale (or underlying hypothesis) for the study, the data are difficult to see and understand, and the interpretation and discussion are superficial.

7. The qRT-PCR study is not well described. For example, it is unclear whether the Ex analysis was done after acute or chronic exercise, which is important but not noted. Also the analyses were noted to be done on duplicate samples but 6 sample symbols shown in the graphs (duplicate biological replica and triplicate on each replica?). There are large variations in the data and it seems that additional sample size would be helpful to support the conclusion. Also, complete qRT-PCR analyses on all knockdowns in all tissue types used for behavioral experiments (only subsets were done) would be needed to make solid conclusions.

8. Discussion on the data obtained using global (TubGS RNAi in the previous paper) versus tissue-specific (this paper) knockdowns of individual octopamine receptors would be helpful.

Minor points:

1. Materials and methods

1.1. The authors note that the behaviors are scored blindly when possible. It is important to note which behaviors are scored blindly and which behaviors are unable to score blindly.

1.2. Information on most fly strains used in the study is missing

1.3. Inconsistent formatting in materials and method for product info (i.e. the city and state information is noted in some but not all)

2. Writing/figures/tables

2.1. Non-italicized Drosophila (P2 L39); Fig S1C, S2A, S2C - genotypes are not italicized

2.2. mislabeling in Fig 2E: it appears hand>OctB1R for both red solid line and red dashed line – missing UN and EX

2.3. In P4 L89-91, it states “Neither heart specific Octβ1R RNAi nor adult fat-body specific RNAi against Octβ3/β1R reduced exercise dependent improvements in endurance (Figures 2E, H).” but Fig 2H depicts B3 knockdown only.

2.4. “Crowded” labeling in figures, especially genotype labels (see fig 4E, 4F, 5E, 5F, etc.)

2.5. In P7 L89-91, it states “…knockdown of OAMB, OctB3R or OctB1R in muscle…” but Fig 6C only depicts OAMB knockdown.

2.6. Statistical analyses are missing in several places (e.g. figure 3)

2.7. Table 1 - mixed symbols (asterisk and minus sign) and letters make the content difficult to understand.

2.8. Table 2 – incomplete and not so informative

Reviewer #3: A previous work by Sujkowski et al. showed that increased activity of octopaminergic neurons is both necessary and sufficient for exercise adaptations in Drosophila. This manuscript is a follow up on that work and here the authors examined the requirements for each of the 4 octopamine receptors in different tissues on the exercise response. They found that each of the octopamine receptors is required for parts of the exercise response in a tissue-specific way and specific to particular phenotypic outputs within the broad suite of exercise responses. They also identify that octopamine signaling in one tissue can affect other tissue phenotypes. While the findings showing the detailed dissection of the input of each octopamine receptor on exercise response is appreciated, the primary concern is that this work is largely descriptive and has not carried things far enough forward, in terms of scope/mechanism and novelty, that would elicit strong general interest from the general readership of the journal here. The study at this stage may be more appropriate for a more specialized journal.

Concerns:

1) Figure 1 – The extent of RNAi knockdown of the 4 octopamine receptors using MHC GS, Hand, and S106 should be determined by QRT-PCR or muscle/heart immunostaining. Even though these RNAi efficiencies have been previously confirmed, as stated.

2) Figure 6 – The impact on autophagy/lysosomal activity by the various tissue knockdowns of the octopamine receptors has to be studied in more detail using more indicators/markers of autophagy/lysosome, not just Lysotracker.

3) All the inhibitory manipulations were performed using genetic knockdowns, not knockouts. Therefore the genetic backgrounds are not “clean” (i.e. there are residual proteins) which could confound the interpretations. For eg., when knocking down one octopamine receptor in one tissue does not affect runspan while knocking down the same/another receptor in another tissue does, is it really because one is dispensable while the other is not, or because one is more depleted than the other?

4) Some mechanistic insights on the tissue non-autonomous effects of cardiac OAMB knockdown would heighten the enthusiasm for the work, especially given that OAMB reduction in the heart produces profound cell-autonomous effects of receptor activation during endurance exercise.

**Have all data underlying the figures and results presented in the manuscript been provided?**

Reviewer #1: Yes

Reviewer #2: Yes

Reviewer #3: Yes

PLOS authors have the option to publish the peer review history of their article (what does this mean?). If published, this will include your full peer review and any attached files.

Reviewer #1: No

Reviewer #2: No

Reviewer #3: No

---

## [Decision Letter · Decision Letter 1]

10 Mar 2020

Dear Dr Wessells,

Thank you very much for submitting your Research Article entitled 'Alpha- and beta-adrenergic octopamine receptors in muscle and heart are required for Drosophila exercise adaptations' to PLOS Genetics. Your manuscript was fully evaluated at the editorial level and by independent peer reviewers. The reviewers appreciated the attention to an important topic but identified some aspects of the manuscript that should be improved.

We therefore ask you to modify the manuscript according to the review recommendations before we can consider your manuscript for acceptance. Your revisions should address the specific points made by each reviewer.

[LINK]

Yours sincerely,

Hua Bai, Ph.D.

Guest Editor

PLOS Genetics

Gregory P. Copenhaver

Editor-in-Chief

PLOS Genetics

Your manuscript PGENETICS-D-19-01309R1 has now been reviewed. Please find enclosed the review comments. Most of the concerns raised by previous reviewers have been adequately addressed, although there are several minor issues remained. In light of these comments, we cannot accept the manuscript for publication. However, we would be interested in reconsidering a revised version that addresses these concerns. Please carefully address the new comments from Reviewer #3 and #4. Please provide thorough discussion on the different phenotypes observed between ubiquitous and tissue-specific KD, and carefully describe the knockdown efficiencies in the text accordingly.

We hope you find the reviewers' comments useful as you decide how to proceed and we are looking forward to your revised manuscript.

Reviewer's Responses to Questions

**Comments to the Authors:**

Reviewer #1: The authors have done an excellent job addressing the concerns I raised. The addition of Figure 7 is helpful. Minor note, in the response to reviewers it is really helpful to provide line numbers of the corrections and new figure numbers instead of “new figure” to help the reviewer quickly identify the change and the text context of the change.

However, congratulations on this exciting paper.

Reviewer #3: Review uploaded as an attachment

Reviewer #4: Summary

The revised manuscript by Sujkowski et. al. extends their investigation on the role of the octopamine (OA) system in exercise adaptations and here, they detail the tissue types where the individual OA receptors function. This rigorous characterization yielded novel insights, especially for the invertebrate exercise field. The sizeable datasets are impressive, logically organized and well presented in the figures. Where the manuscript falls short, in my opinion and as the previous reviewers pointed out, is its limited scope (~ solely focused on the identification of receptor functional roles) and the lack of information on downstream mechanisms. The authors point out that this notion is beyond the scope of this manuscript, and that their finding that the OA receptors cannot functionally substitute for each other is an important novel mechanistic insight, is sufficient for publication. With all of this in mind, there are several points that should be clarified prior to consideration for publication (listed below).

Comments

The authors have addressed almost all of the major points brought up by the three reviewers, however, there is one comment that in my opinion was insufficiently addressed. On reviewer 2 comment 8, the authors should elaborate further on the discrepancies between the phenotypes observed using global KD (Cell reports 2017) versus tissue specific KD (this revised manuscript) of individual OA receptors. For example, address questions including:

A. Why ubiquitous Octb3R KD versus fat body Octb3R KD yielded different results in the lysosomal activity? This is especially important since it is one of the major findings attributed to Octb3R function in exercise adaptation.

B. Why ubiquitous Octb1R KD versus either muscle or fat body Octb1R KD have different effects on endurance, as measured by runspan? The authors only address flight.

C. Why ubiquitous Octb2R KD affects all exercise adaptations (Cell reports 2017) yet in this was not the case for tissue specific Octb2R KD? Especially address the discrepancy on flight.

This section in the discussion can touch on the the known signaling pathways that the different OA receptors trigger (for example: OAMB—Ca2+--CaMKII) that may help explain discrepancies in phenotype expression and begin to describe possible molecular mechanisms downstream of OA receptors affected by exercise. This, in part, may address the similar concerns expressed by the other reviewers.

The experiment using with OA feeding in the flies with OA receptor KD (Octb2R and Octb3R) in muscle are great. These results indeed facilitated the identification of receptor requirement or functional substitution for exercise responses. The data shown is on the Octb2R and Octb3R in muscle tissue, a great proof of concept. However, in lines 228-231, this statement in my opinion, is too bold. This generalized statement can be easily misinterpreted to encompass all OA receptors in different tissue types (e.g. OAMB and Octb1R). Unless, there are OA feeding data on all individual OA receptor KDs, the authors should amend this statement.

In line with the previous comment, the authors argue the novel mechanism that “…OA receptors cannot substitute for each other, even in the same tissue…” is indeed true for exercise adaptations. This concept, however, has been demonstrated in the female reproductive system epithelium in Lim et. al., 2014 (ref 41), where the OAMB could not functionally substitute for Octb2R in the to restore fecundity yet OAMB can partially substitute for Octb2R to restore the ovulation phenotype of the Octb2R mutant females. Both OAMB and Octb2R are expressed in the female oviduct epithelium and are important for female reproduction. The authors may reference this to support their findings and hypothesis on the similar phenomenon observed for exercise adaptations.

Are the specific OA neurons required for exercise adaptations known or mapped? If so, does it involve all or subset (e.g. Tdc2+, VUM, VPM or APL)? Do these OA neurons send axonal projections to the different tissue types (muscle, heart or fat body) tested? Alternatively, is OA released into the fly circulation? The authors can add this information in the discussion.

In lines 186-188, the statement “…trends toward lower levels…”, in my opinion, is misleading. There is no statistics information presented to suggest the “lowering” trend. The authors should rephrase this sentence.

Lines 197-198 state, “summary of OA-feeding+exercise results is in Table 1”, there is no Table 1 either the manuscript doc, pdf, or supplemental doc. This should be clarified.

Lines 268-269 state, “OAMB, the only Drosophila α-adrenergic OA-receptor…” is incorrect. There is another one, the Octα2R that the authors reference – Qi et. al. 2017 (ref 48). This sentence needs to be revised.

In line 681, missing close parenthesis after “(K”.

The statistics information for the cardiac pacing experiment is missing in the methods section.

The information for S106 GS is missing from the fly stocks section.

The labeling in Figure 4G is not easy to follow especially the “WT(-)(+)” on top of the blot image – an explanation in the figure legend would be helpful.

The Figure 7 legend states, “…Cell non-autonomous effects…indicated by colored arrows.”, but there three sets of colored arrows in the summary: (1) denoting increased OA, (2) arrows denoting OA binding to different OA receptors and (3) cell non-autonomous effects. This should be clarified.

The first two sentences of the abstract are identical to the authors’ previous publication on Cell reports 2017. The authors should rephrase/reword these sentences.

**Have all data underlying the figures and results presented in the manuscript been provided?**

Reviewer #1: Yes

Reviewer #3: Yes

Reviewer #4: Yes

PLOS authors have the option to publish the peer review history of their article (what does this mean?). If published, this will include your full peer review and any attached files.

Reviewer #1: No

Reviewer #3: No

Reviewer #4: No

---

## [Decision Letter · Decision Letter 2]

15 Apr 2020

Dear Dr Wessells,

We are pleased to inform you that your manuscript entitled "Alpha- and beta-adrenergic octopamine receptors in muscle and heart are required for Drosophila exercise adaptations" has been editorially accepted for publication in PLOS Genetics. Congratulations!

Yours sincerely,

Hua Bai, Ph.D.

Guest Editor

PLOS Genetics

Gregory P. Copenhaver

Editor-in-Chief

PLOS Genetics

Comments from the reviewers (if applicable):

Reviewer's Responses to Questions

**Comments to the Authors:**

Reviewer #1: My concerns have been well-addressed.

Reviewer #3: The authors have responded satisfactorily to my concerns.

Reviewer #4: Summary

Sujkowski et. al. expands our understanding on the octopamine system’s role in exercise adaptations by detailing the individual octopamine receptors' function in different exercise-relevant tissue types. The study serves as a significant entry point for further investigation on the comprehensive mechanism(s) governing exercise response in Drosophila.

Comments

The authors have addressed almost all my concerns and I would like to congratulate the authors on their exciting manuscript. The only thing that remains missing, as I could still not locate it in the submission, is the Table 1. Once this error is corrected, in my opinion, the manuscript should be ready for publication.

**Have all data underlying the figures and results presented in the manuscript been provided?**

Reviewer #1: Yes

Reviewer #3: Yes

Reviewer #4: Yes

PLOS authors have the option to publish the peer review history of their article (what does this mean?). If published, this will include your full peer review and any attached files.

Reviewer #1: No

Reviewer #3: No

Reviewer #4: No

**Data Deposition**

http://datadryad.org/submit?journalID=pgenetics&manu=PGENETICS-D-19-01309R2

**Press Queries**

---

## [Editor Report · Acceptance letter]

9 Jun 2020

PGENETICS-D-19-01309R2 

Alpha- and beta-adrenergic octopamine receptors in muscle and heart are required for Drosophila exercise adaptations 

Dear Dr Wessells, 

We are pleased to inform you that your manuscript entitled "Alpha- and beta-adrenergic octopamine receptors in muscle and heart are required for Drosophila exercise adaptations" has been formally accepted for publication in PLOS Genetics! Your manuscript is now with our production department and you will be notified of the publication date in due course.

With kind regards,

Kaitlin Butler

PLOS Genetics

On behalf of:
